# Constrained Diffusers for Safe Planning and Control

**Jichen Zhang    Liqun Zhao    Antonis Papachristodoulou    Jack Umenberger**
University of Oxford
{jichen.zhang, liqun.zhao, antonis, jack.umenberger}@eng.ox.ac.uk

## Abstract

Diffusion models have shown remarkable potential in planning and control tasks due to their ability to represent multimodal distributions over actions and trajectories. However, ensuring safety under constraints remains a critical challenge for diffusion models. This paper proposes Constrained Diffusers, an extended framework for planning and control that incorporates distribution-level constraints into pretrained diffusion models without retraining or architectural modifications. Inspired by constrained optimization, we apply a constrained Langevin sampling method for the reverse diffusion process that jointly optimizes the trajectory and achieves constraint satisfaction through three iterative algorithms: projected method, primal-dual method and augmented Lagrangian method. In addition, we incorporate discrete control barrier functions as constraints for constrained diffusers to guarantee safety in online implementation, following a receding-horizon control that we generate a short-horizon plan and execute only the first action before replanning. Experiments in Maze2D, locomotion, and PyBullet ball running tasks demonstrate that our proposed methods achieve constraint satisfaction with less computation time, and are competitive with existing methods in environments with static and time-varying constraints. The implementation can be found here.

## 1 Introduction

In recent years, diffusion models that learn data distributions by gradually adding noise and then reversing the process have achieved remarkable success in image generation [1]. The forward process corrupts expert data into noise via predefined Gaussian steps, while the reverse process trains a neural network to iteratively denoise, reconstructing the original distribution from random noise [2]. This success has naturally extended to policy representation in planning and control, surpassing traditional imitation learning by denoising trajectories for planning [3] and modeling policies as return-conditional diffusion models to obtain optimal trajectories [4].

Despite these advances, deploying diffusion-based planning policies on real physical systems raises safety challenges, as such systems operate under strict constraints. Recent works incorporate constraints as conditions within generative models—e.g., classifier-guided [5] and classifier-free [6] methods. Although effective in image or text generation, their applications in planning/control still lead to substantial violations under nonconvex, time-varying constraints and often require retraining for specific constraints [3, 4, 7, 8]. Other efforts enforce safety with Control Barrier Functions (CBFs) at each reverse step [9], but this involves solving a Quadratic Program (QP) per diffusion step, incurring heavy computation. Projection-based approaches instead map sampled points to the safe set [10, 11], effectively guaranteeing strict constraint satisfaction for samples during the diffusion process. However, they typically focus on pointwise constraints rather than general expectation constraints on distributions. Furthermore, the hard constraint projection can potentially disrupt trajectory smoothness, posing challenges for safe online implementation.

We propose *Constrained Diffusers* for planning and control, formulating diffusion generation with constraints on distributions through the lens of Stochastic Gradient Langevin Dynamics (SGLD)

39th Conference on Neural Information Processing Systems (NeurIPS 2025).

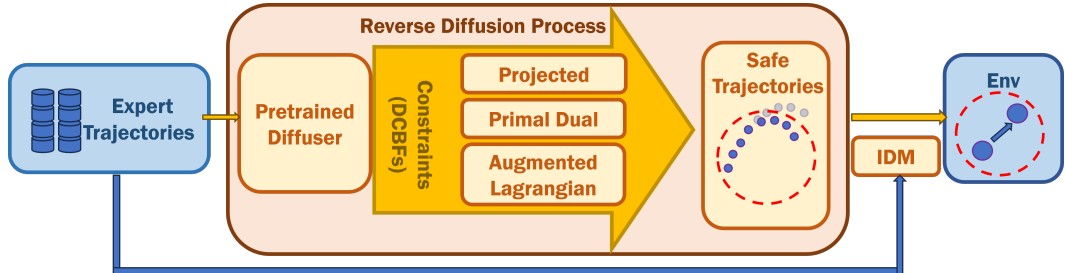

Figure 1: Overall Framework. We use expert trajectories to train the diffuser, apply constrained sampling methods for DCBF constraints in the reverse sampling process to get safe trajectories and finally use an inverse dynamics model to obtain actions, ensuring the safety in online implementation. We follow a receding-horizon scheme that only the first action is executed before replanning.

[12, 13]. We interpret generation as constrained sampling, viewing the iterations as stochastic gradient descent, and integrate constraints using projected, primal–dual, and augmented Lagrangian methods (ALM). To keep system states within the safe set, we impose discrete CBF (DCBF) constraints during the reverse diffusion process. This coupling allows reverse steps to optimize trajectories while enforcing safety without solving QPs for DCBF, and avoids retraining when a pre-trained diffusion model exists for the task. Building on [14, 15, 16, 17, 18], we analyze convergence under mild conditions and provide constraint-satisfaction guarantees. We use an inverse dynamics model, trained on datasets, to map states to actions and execute control in a receding horizon scheme. At each step, we replan over a short horizon, execute the first action, update with the new observation, and repeat, mirroring Model Predictive Control (MPC) and mitigating errors by replanning.

The main contributions of our work are summarized as follows:

- We propose Constrained Diffusers, an extended framework for planning and control that enforces distribution-level constraints for trajectories generated by pre-trained diffusion models, modifying the reverse diffusion process with constrained sampling methods—Projected [10], a novel Primal-Dual method, and Augmented Lagrangian—thus avoiding retraining at deployment. We further introduce DCBF on physical trajectories to ensure safe receding-horizon execution that executes only the first action in the plan, then repeat, leveraging the iterative diffusion process for efficient constraint satisfaction.

- We demonstrate the effectiveness of our proposed methods in Maze2D, locomotion, PyBullet ball running tasks and manipulation tasks in the Appendix showing that our method effectively enforces constraints with competitive performance and computational efficiency.

## 2   Related Work

**Diffusion Models and Policy Representation** Diffusion models [1, 2] have demonstrated significant potential in image [5, 6] and text generation [19, 20]. They gradually add noise to the data, transforming them into a random distribution over time [12, 21] and learn to iteratively denoise the data by reversing the forward diffusion steps [22]. This success extended to planning tasks [3, 4], outperforming traditional imitation learning methods through the ability to model complex distributions [23, 7]. Recent literature focuses on trajectory optimization through diffusion models [24, 25] while they require interactions with environments in the optimization process. Some researchers have also investigated how diffusion can be combined with reinforcement learning to guide strategy improvement [8]. However, they did not handle constraints in the trajectory generation and online implementation through diffusion.

**Constrained Sampling** Constrained sampling methods have been developed for support constraints in the standard sampling process [26]. Projected Langevin Monte Carlo extends the algorithm to compactly supported measures through a projection step when sampling from a log-concave distribution [15]. Primal-dual methods simultaneously sample from the target distribution and constrain it through gradient descent-ascent dynamics in the Wasserstein space [16, 17]. In addition, some Langevin Monte Carlo methods based on mirror maps [27], barriers [28], and penalties [29]

have also demonstrated significant advantages in constrained sampling. However, they do not make connections with diffusion models and do not extend them to nonconvex settings.

**Safe Reinforcement Learning and Control Barrier Functions** Safe control is a central research area in robotics and autonomous systems [30]. Traditional model-based methods—Hamilton–Jacobi reachability [31] and Model Predictive Control [32]—offer rigorous guarantees. Safety-reinforcement learning typically formulates a Constrained Markov Decision Process, optimizing expected cumulative costs with trajectory-level constraints [33, 34, 35]; safe exploration adds safety layers to adjust actions [36], and other algorithms strengthen policy robustness to safety risks [37]. Beyond safe RL, constrained sampling methods include task-constrained RRT [38], while overall complexity limits broad applicability across robotics [39]. Control Barrier Functions (CBFs) are widely used for safe control [40, 41]; recent work embeds CBFs in diffusion to ensure generated trajectories avoid obstacles [9]. [42] proposes BarrierNet, a differentiable layer that learns CBF parameters for end-to-end training of less conservative policies. Other studies tackle safety in imitation learning and uncertainty by solving QPs, which is time-consuming [43, 44, 45].

Compared to classical planners, we use diffusion models as planners for their ability to represent complex, multimodal trajectory distributions—well-suited when objectives are unknown and policies come from expert demonstrations. Compared to *SafeDiffuser* [9], which interprets each denoising step as a control action over the diffusion horizon and applies CBFs at intermediate steps, and *Projected Diffusion* [10], which works on constraints for samples, our PD and ALM treat diffusion as a constrained optimization problem on distributions, enforce DCBFs along the entire trajectory in the physical world, and use efficient first-order updates to ensure safety—thereby avoiding costly per-step QP solves. Detailed comparisons with Projected Diffusion and summarized table on comparisons with other literature can be found in Appendix A.

## 3 Background

### 3.1 Diffusion Models and Langevin Dynamics

**Diffusion models** are a type of generative model inspired by principles of thermodynamics and work by simulating a two-step process. The forward process perturbs data $x_0 \sim q(x_0)$ by gradually adding noise over diffusion time $t \in [0, T]$, and the reverse process reconstructs the target data distribution from a simple prior distribution. These processes can be formulated as the following SDEs [21]:

$$dx_t = f(x_t, t)dt + g(t)dW_t, \quad dx_t = [f(x_t, t) - g(t)^2 \nabla_{x_t} \log p_t(x_t)]dt + g(t)d\bar{W}_t, \quad (1)$$

where $W_t$ and $\bar{W}_t$ are standard Wiener processes running forward and backward in time, $f(x_t, t)$ and $g(t)$ are drift and diffusion coefficients chosen such that $p_T(x_T)$ approximates a simple prior, e.g., $\mathcal{N}(0, I)$, and $\nabla_{x_t} \log p_t(x_t)$ is the score function of the marginal distribution $p_t(x_t)$.

**Langevin Dynamics** is a method that leverages gradient information and noise to sample from a given distribution, resembling simulating the diffusion reverse SDE [21] starting from $x_T \sim p_T$ and evolving backward to $t = 0$. Standard Langevin Monte Carlo (LMC) aims to sample from a target distribution $p(x) \propto e^{-f(x)}$ using the update rule derived from the Langevin SDE:

$$dx(t) = -\nabla f(x(t))dt + \sqrt{2}dW(t). \quad (2)$$

In practice, due to the challenges in directly computing the path of SDE, we often use a discrete-time approximation, Stochastic Gradient Langevin Dynamics (SGLD):

$$x_{t+1} = x_t + \frac{\eta}{2} \nabla_x \log p(x_t) + \sqrt{\eta} z_t \quad (3)$$

where $\eta$ is the step size, and $z_t \sim \mathcal{N}(0, I)$ is standard Gaussian noise. By iterating this update rule with an appropriate step size $\eta$, the sequence $x_t$ can be considered as samples drawn from $p(x)$.

**Denoising Diffusion Probabilistic Model (DDPM)** can be connected to a Langevin-like update rule through its iterative denoising process. In DDPM, the forward process gradually adds Gaussian noise to data over $T$ steps, defined by $q(x_t | x_{t-1}) = \mathcal{N}(x_t; \sqrt{1 - \beta_t} x_{t-1}, \beta_t \mathbf{I})$, where $\beta_t$ controls the noise schedule. The reverse process, $p_\theta(x_{t-1} | x_t)$, approximates the data distribution by learning the noise via a neural network, enabling the following reverse update:

$$x_{t-1} = \frac{1}{\sqrt{1 - \beta_t}} \left( x_t - \frac{\beta_t}{\sqrt{1 - \bar{\alpha}_t}} \epsilon_\theta(x_t, t) \right) + \sqrt{\beta_t} z, \quad (4)$$

where $\epsilon_\theta$ is the learned noise estimate, $\bar{\alpha}_t = \prod_{s=1}^{t}(1 - \beta_s)$ and $z_t \sim \mathcal{N}(0, I)$ is standard Gaussian noise. This reverse step resembles Langevin dynamics, where the score function $\nabla_{x_t} \log p(x_t)$ is implicitly approximated by the denoising model's prediction of the noise component. According to [46], we can rewrite the score function with respect to the noise term by combining Tweedie's Formula with this reparameterization:

$$\nabla_{x_t} \log p(x_t) = -\frac{\epsilon_\theta(x_t, t)}{\sqrt{1 - \bar{\alpha}_t}} = -\frac{x_t - \sqrt{\bar{\alpha}_t}x_0}{(1 - \bar{\alpha}_t)}. \tag{5}$$

This reformulates the DDPM reverse process update into a Langevin Sampling process in terms of the score function $\nabla_{x_t} \log p(x_t)$ instead of the noise estimate $\epsilon_\theta(x_t, t)$.

$$x_{t-1} = x_t + \frac{\beta_t}{2}\nabla_{x_t} \log p(x_t) + \sqrt{\beta_t}z. \tag{6}$$

This formulation bridges the DDPM reverse process to stochastic gradient-based Langevin dynamics, aligning denoising with probabilistic inference, forming the basis for leveraging optimization techniques within the sampling process. To ensure consistency in the time scale, we assume the iterative process proceeds from $T$ to $0$ as $T \to \infty$ in this paper.

## 3.2 Constrained Sampling

In practical applications, it is crucial to generate samples that satisfy specific constraints or physical laws. This motivates the problem of *constrained sampling*, where the goal is to sample from a distribution $q$ that is "close" to a reference distribution $p$ (e.g., the distribution implicitly defined by expert data) while satisfying certain constraints. As formulated by [16], this can be posed as an optimization problem in the space of probability measures:

$$q^* = \underset{q \in \mathcal{P}_2(\mathcal{R}^d)}{\arg\min} \quad \mathrm{KL}(q||p) \quad \text{s.t.} \quad \mathbb{E}_{x \sim q}[g(x)] \leq 0, \tag{7}$$

where $g(\cdot)$ represent inequality constraint functions and $\mathrm{KL}(q||p)$ measures the divergence from the reference distribution $p$.

For the task we aim to solve, we do not assume constraints are convex. Therefore, our approach involves adapting existing methods and extending them to non-convex settings, while proving the effectiveness of the proposed algorithm in experiments and giving theoretical analysis.

## 3.3 Discrete Control Barrier Functions

To enforce safety constraints in dynamical systems, Control Barrier Functions (CBFs) are designed to ensure forward invariance of a desired safe set $\mathcal{C}$, which is typically defined as the superlevel set of a continuously differentiable function $h(x)$, i.e., $\mathcal{C} = \{x \in \mathbb{R}^d : h(x) \geq 0\}$ [40].

For a general discrete-time deterministic system $x^{\tau+1} = f_d(x^\tau, u^\tau)$, where $x^\tau \in \mathbb{R}^d$ is the system state at physical system time step $\tau$, $u_\tau \in \mathcal{U} \subset \mathbb{R}^m$ is the control input, and $f_d$ denotes the discrete-time system dynamics, a function $h(x)$ is a Discrete Control Barrier Function (DCBF) if there exists an extended class $\mathcal{K}$ function $\alpha$ such that for all $x \in \mathcal{C}$[41]:

$$h(f_d(x^\tau, u^\tau)) - h(x^\tau) \geq -\alpha(h(x^\tau)). \tag{8}$$

This condition ensures that the decrease in $h(x)$ between discrete time steps is bounded, thus preserving the forward invariance of the safe set $\mathcal{C}$. In practice, this often involves solving a quadratic programme (QP) at each time step to find a control input $u$ that satisfies the CBF condition while minimally deviating from a nominal control policy.

## 4 Problem Statement

The generative power of diffusion models extends to sequential decision-making problems, leveraging the models to represent distributions over trajectories or action sequences. Specifically, these models are trained to model the distribution over expert demonstrations by corrupting trajectories $(x^0, \ldots, x^\mathcal{T})$ with noise in a forward diffusion process and learning to reverse this corruption during training [3]. Despite the demonstrated success of diffusion models in trajectory planning and control,

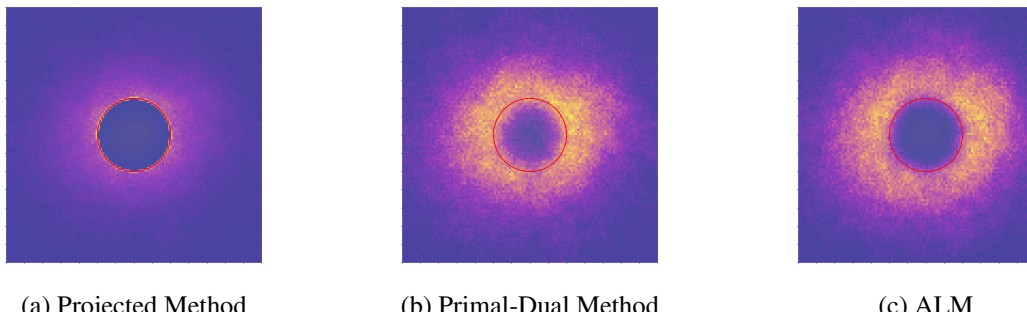

| (a) Projected Method | (b) Primal-Dual Method | (c) ALM |

Figure 2: Constrained sampling trajectories through (a) projected, (b) primal-dual and (c) augmented Lagrangian Method. The goal is to use Langevin sampling methods to sample from a 2-dimensional Gaussian distribution with the constraint $x^2 + y^2 \geq r^2$. Yellow regions show the high density of Langevin sampling trajectory distributions. Trajectories of the projected method mostly concentrate on the constraint boundary while the other two are more distributed.

their practical deployment in safety-critical systems faces a fundamental challenge: enforcing dynamic constraints during the sampling process without costly retraining or architectural modifications.

To address this, we propose solving the constrained generation problem by formulating it as the following optimization problem: given a pretrained diffusion model defining an implicit distribution $p(x^{0:\mathcal{T}})$ over trajectories, and constraint functions $g(x^{0:\mathcal{T}})$, where $\mathcal{T}$ is physical system time step, we design an alternative reverse process that produces a "similar" trajectory distribution $q^*$ satisfying:

$$q^* = \underset{q \in \mathcal{P}_2(\mathcal{R}^d)}{\arg\min} \quad \mathrm{KL}(q||p) \quad \text{s.t.} \quad \mathbb{E}_{x^{0:\mathcal{T}} \sim q}[g(x^{0:\mathcal{T}})] \leq 0, \tag{9}$$

while remaining computationally tractable. We assume access to the differentiable constraints and their gradients. For notational brevity, we omit the superscript $0 : \mathcal{T}$ when possible.

*Remark* 1. For constraints on samples rather than expectations under distributions, we could formulate the constraints as $\mathbb{E}_{x^{0:\mathcal{T}} \sim q}[[g(x^{0:\mathcal{T}})]_+] \leq 0$, which ensures the probability of any constraint violation is exactly zero under $q$. For cases where the constraints need to be differentiable, we can use $\mathbb{E}_{x^{0:\mathcal{T}} \sim q}\left[\frac{g(x^{0:\mathcal{T}})}{1+e^{-g(x^{0:\mathcal{T}})}}\right] \leq 0$ as a smooth approximation of $[g(x^{0:\mathcal{T}})]_+$.

In the context of control systems, we implement these constraints using DCBF to ensure safety. The DCBF conditions are treated as the specific constraint imposed on the system's actuation dynamics within our general formulation in (9). This approach provides a plug-and-play mechanism to enforce constraints directly into the reverse diffusion process, as illustrated in our framework in Figure 1.

## 5 Constrained Diffusers

In this section, we present three methods for handling constraints during reverse diffusion sampling: Projected methods, Primal-Dual methods, and Augmented Lagrangian methods. Visualization of the modified reverse process can be found in Appendix N.

### 5.1 Projected Diffusion Sampling

This approach leverages Projected Gradient Descent (PGD) to enforce trajectory constraints during the diffusion denoising process, extended from the work [10], which has shown that projected diffusion model can be successfully applied to some planning tasks, with analysis on the convergence and applications to the control tasks. At each denoising diffusion time step $t$ of the diffusion process, we modify the standard update rule with constraint projection as follows,

$$x_{t-1} = \Pi_{\mathcal{C}} \left( x_t + \frac{\beta_t}{2} \nabla_{x_t} \log p(x_t) + \sqrt{\beta_t} z \right) \tag{10}$$

where $\Pi_{\mathcal{C}}$ denotes the projection operator onto the constraint set $\mathcal{C}$ defined in (9). The projection operator $\Pi_{\mathcal{C}}$ solves the constrained optimization problem:

$$\Pi_{\mathcal{C}}(z) = \underset{x \in \mathcal{R}^d}{\arg\min} \|x - z\|^2 \quad \text{s.t.} \quad x \in \mathcal{C} \tag{11}$$

Through these updates, the samples will converge to stationary points. Detailed analysis to the statement can be found in Appendix I. This method applies for constraints for samples and the constraints are met at every diffusion step.

## 5.2 Primal-Dual Methods (PD)

This approach addresses the constrained optimization problem (9) by solving the following problem:

$$\max_{\lambda \geq 0} \min_{q} \left\{ L(q, \lambda) = \text{KL}(q||p) + \lambda^{\mathsf{T}} \mathbb{E}_{x \sim q}[g(x)] \right\}. \tag{12}$$

where $\lambda$ is the corresponding Lagrange multiplier.

**Definition 5.1.** A local saddle point of $L(q, \lambda)$ is a point $(q^*, \lambda^*)$ such that for some $r > 0$, $\forall q \in \mathcal{P}_2(\mathcal{R}^d) \cap \mathcal{B}_{q^*}(r)$ and $\forall \lambda \geq 0$, we have

$$L(q, \lambda^*) \geq L(q^*, \lambda^*) \geq L(q^*, \lambda),$$

where $\mathcal{B}_{q^*}(r)$ is a ball centered at $q^*$ with radius $r > 0$ under the 2-Wasserstein metric.

According to [47, 48], a local saddle point for the maxmin problem is a local optimal solution to the primal constrained optimization problem (9). The optimal constrained distribution $q^*$ can be characterized as a tilted version of the original distribution $p(x) \propto e^{-f(x)}$, given by $q^*(x) \propto e^{-(f(x) + \lambda^T g(x))}$. Sampling from $q^*$ can then be achieved by modifying the LMC dynamics to target this tilted distribution. In practice, we solve the maxmin problem by alternatively updating $x$ and $\lambda$: The primal update is to incorporate gradients related to the constraint functions; the dual update is to ascend in $\lambda$ using the gradients of $L(q, \lambda)$ w.r.t. $\lambda$, i.e.,

$$x_{t-1} = x_t + \frac{\beta_t}{2} (\nabla_{x_t} \log p(x_t) - \lambda_t^T \nabla_{x_t} g(x_t)) + \sqrt{\beta_t} z_t, \tag{13}$$

$$\lambda_{t-1} = [\lambda_t + \eta_\lambda \mathbb{E}_{x_t \sim q}[g(x_t)]]_+. \tag{14}$$

where $[\cdot]_+ = \max(0, \cdot)$ denotes projection onto the non-negative orthant, and $\eta_\lambda$ are step sizes.

**Theorem 5.1.** *Under mild conditions, the sequence of updates* (13) *and* (14) *converges almost surely to a local saddle point, i.e. a local optimal solution to the constrained optimization problem* (9), *as T goes to infinity.*

The detailed proof of the theorem can be found in the Appendix J. In other words, under the proposed update strategy, when a score function already exists, constraint satisfaction can be achieved by their gradients with respect to the samples, eliminating the need for solving a QP.

## 5.3 Augmented Lagrangian Methods (ALM)

This approach enhances constraint handling by introducing a quadratic penalty term into the Lagrangian relaxation which enhances numerical stability[49, 50]. To handle inequality constraints $\mathbb{E}_{x \sim q}[g(x)] \leq 0$, we introduce slack variables $s \geq 0$ and let $\mathbb{E}_{x \sim q}[g(x)] + s = 0$. Then, we reformulate the constrained optimization problem (9) using the augmented Lagrangian:

$$L_A(q, \lambda, s) = \text{KL}(q||p) + \lambda^{\mathsf{T}}[\mathbb{E}_{x_t \sim q}[g(x_t)] + s] + \frac{\rho}{2} \|\mathbb{E}_{x_t \sim q}[g(x_t)] + s\|^2 \tag{15}$$

The slack variable $s$ enables exact constraint satisfaction while maintaining differentiability. The update scheme features three components:

$$x_{t-1} = x_t + \frac{\beta_t}{2} \left[ \nabla_{x_t} \log p(x_t) - (\lambda_t + \rho_t (\mathbb{E}_{x_t \sim q}[g(x_t)] + s_t))^T \nabla_{x_t} g(x_t) \right] + \sqrt{\beta_t} z_t, \tag{16}$$

$$s_t = [-\mathbb{E}_{x_t \sim q}[g(x_t)] - \lambda_t / \rho_t]_+, \text{(Slack Update)} \tag{17}$$

$$\lambda_{t-1} = \lambda_t + \rho_t (\mathbb{E}_{x_t \sim q}[g(x_t)] + s_t), \text{(Dual Update)} \tag{18}$$

To obtain $\rho \to \infty$, we update the penalty term $\rho_{t-1} = c \cdot \rho_t (c > 1)$. These updates will finally converge to the local optimal solution of (9) under certain conditions. Detailed analysis of the statement can be found in Appendix K. Compared to the primal-dual method, this method provides better constraint satisfaction for inequality-constrained diffusion processes.

# 6  Online Safety Implementation

To ensure that the trajectories produced by our model remain safe during online implementation with actions feeding into environments, we incorporate two key components: discrete control barrier functions (DCBFs) and inverse dynamics model based on the following two assumptions: 1) that an inverse dynamics model exists for the system, although it is not accessible in a closed form derived from first-principle physics, 2) that there are no constraints on the inputs.

## 6.1  Discrete Control Barrier Function Constraints in Physical Environments

To enhance safety during the implementation stage, we incorporate discrete control barrier functions in our reverse diffusion process. In practice, this condition is typically enforced through a QP at each time step. Since our approach directly generates trajectories, we could directly consider safety constraints between consecutive states $x^\tau$ and $x^{\tau+1}$ at the trajectory level. Specifically, we guarantee that consecutive states in our generated trajectories satisfy the safety condition by rewriting (8):

$$h(x^{\tau+1}) \geq (1-\alpha)h(x^\tau), \quad \forall \tau = 0, \ldots, \mathcal{T}, \tag{19}$$

here we use a proportional function to replace extended class $\mathcal{K}$ function where $\alpha \in (0,1]$ is the coefficient that determines how aggressively the system state is required to remain within or approach the safe set. By enforcing this condition directly on the state trajectory, we ensure that the system trajectory evolves safely without explicitly computing control actions during the generation process. The detailed update scheme for DCBF constraints can be found in Appendix L.

## 6.2  Inverse Dynamics Model

When applying the proposed methods at implementation stage in control tasks, the diffuser generates sequences of states. To guarantee the consistency of state and action, we employ an inverse dynamics model (IDM), denoted by $u^\tau = \text{IDM}(x^\tau, x^{\tau+1})$, which predicts the action $u$ required to transition from state $x^\tau$ to state $x^{\tau+1}$ after obtaining the constrained state through the proposed algorithms. This ensures that the action executed corresponds to the constraint-satisfying state achieved after the modified reverse diffusion step. To address the potential inconsistency that the mapping from a state transition to a control input may not be one-to-one in systems with redundancy, our approach implements IDM using a deterministic neural network, which provides a unique action output. Furthermore, the IDM is applied on well-behaved transitions whose distribution closely matches the expert distribution, which further reduces the risk of ambiguity. More clarifications can be found in Appendix M.

The pseudo-code of the complete algorithm is shown below.

---
**Algorithm 1** Constrained Diffusers (use Primal-Dual as an example)

---
**Input parameters:** Expert data, constraints $g(x)$, variance $\beta$, dual step $\eta_\lambda$, initial Lagrangian multiplier $\lambda$

1: *Training*: Use the expert data to train the score function and IDM.
2: **For** $\tau = 0, ..., \mathcal{T}$: // environment timesteps
3:     Initialize $x_T \sim \mathcal{N}(0, \mathbf{I})$.
4:     *Planning*:
5:     **For** $t = T, ..., 0$: // diffusion timesteps
6:         Update $x_t$ based on (13) with DCBFs as constraints.
7:         Update $\lambda_t$ based on (14)
8:     **end for**
9:     **return** $x_0$ **for planning**
10:     *Control*:
11:     $u^\tau = \text{IDM}(x_0^0, x_0^1)$
12:     Feed $u^\tau$ into environment.
13: **end for**

---

Note that our algorithm involves two time horizons: the diffusion process horizon denoted by $t$ and $T$, and the physical environment horizon denoted by $\tau$ and $\mathcal{T}$. $x_0$ represents the entire clean trajectory

over the planning horizon $\mathcal{T}$ at diffusion timestep 0. $x_t$ denotes the noisy version of this trajectory at diffusion step $t$. The notation $x_0^\tau$ refers to the $\tau$-th state in $x_0$, where $x_0^0$ is the current agent state, and $x_0^1$ is the next planned state. During inference, the agent first observes its current state, which becomes $x_0^0$. Then a full reverse diffusion process is run from noise to generate $x_0$. The constrained sampling method ensures safety state constraints are satisfied. The IDM computes the control $a^\tau$ for the safe state transition $x_0^0 \rightarrow x_0^1$ which is executed in the environment. The remaining trajectory is discarded. After the environment updates to the next state, the process repeats from Step 1. Thus, the policy predicts a full safe trajectory at each time step but executes only the first action.

## 7    Experiments

We evaluate *Constrained Diffusers* on Maze2D planning, locomotion, manipulation, and PyBullet ball-running tasks. Our experiments address four questions: (i) can our methods enforce trajectory constraints *without retraining*? (ii) how do they compare to existing techniques in constraint satisfaction, task success, and computational efficiency? (iii) can they adapt to time-varying constraints during deployment? and (iv) what are the strengths and weaknesses of the proposed approaches integrated within diffusion sampling? Detailed settings are in Appendix C and further experiments on scalability to high-dimensional tasks can be found in Appendix D.

### 7.1    Constraint Satisfaction in Maze2D Planning tasks

First, we evaluate the constrained diffusers on trajectory planning tasks in two Maze2D environments: **Maze2D-umaze** and **Maze2D-large**. We set up obstacles in these environments (Figure 3), requiring the agent's planned trajectory to reach the goal while avoiding these obstacles.    Obstacles are

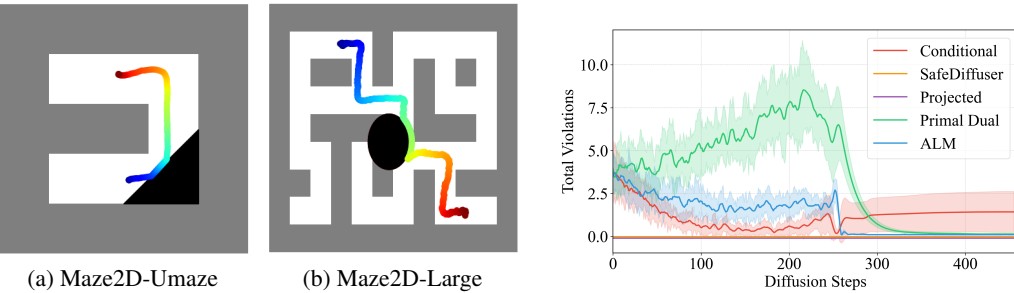

(a) Maze2D-Umaze          (b) Maze2D-Large

Figure 3: Trajectories generated by Constrained Diffuser in Maze2D Environment with Black obstacles. The blue part shows the start of the trajectories and the red part shows the target. We observe that the trajectories avoid the obstacles and reach the goal

Figure 4:  Convergence Results for algorithms in Maze2D. Proposed methods can satisfy constraints. ALM shows better convergence and projected methods satisfy constraints throughout reverse diffusion process.

expressed as linear or quadratic inequalities, which we explicitly embed into the trajectory generation process. We compare against three baselines: *Diffuser* (vanilla trajectory diffusion), *Conditional Diffuser* [4] (constraint-conditioned generation), and *SafeDiffuser* [9] (CBF-guided denoising). Table 1 reports, for both environments, the Euclidean distance to the original trajectory, the total constraint violations $\Sigma_{\tau=0}^{\mathcal{T}} [g(x^\tau)]_+$, and computation time per diffusion step.

From Table 1, conditional diffusion struggles to handle constraints in environments. Both SafeDiffuser and the Projected method can strictly satisfy all constraints, achieving zero violations, because they solve a QP problem at every diffusion step. This strict enforcement, however, comes at a substantial computational cost. In contrast, PD and ALM use numerical iterative updates, which introduce minor violations in the planning phase but still reduce constraint violations by over 99% compared to vanilla Diffuser, while only increasing computation time by approximately 10%. ALM adds a squared constraint penalty, making it slightly more effective in violation reduction than PD at the cost of 10% additional computation, which is still only 1/10 of the closed-form method and 1/100 of QP-based methods. This result demonstrates the ability of our proposed approach: being nearly consistent with expert behavior while flexibly handling constraints to ensure safety, all without retraining.

Table 1: Performance comparison for Maze2D tasks. We can find that our algorithm nearly satisfies constraints and reduce computational time, compared to baseline algorithms. We report the mean and the standard error over 10 random environments.

| Env | Algorithm | Distance | Total Violations | Time(s) |
|---|---|---|---|---|
| maze2d-umaze | Diffuser[3] | 0.00 | 7.316 ± 0.884 | 0.0016 |
| | Conditional[4] | 23.96 ± 2.14 | 1.109 ± 0.834 | 0.0017 |
| | SafeDiffuser[9] | 42.81 ± 8.66 | **0.000 ± 0.000** | 0.1208 |
| | Projected[10] | 10.24 ± 1.39 | **0.000 ± 0.000** | 0.0107 |
| | Primal-Dual | 9.78 ± 1.14 | 0.013 ± 0.032 | **0.0017** |
| | ALM | 9.81 ± 1.65 | 0.002 ± 0.002 | 0.0018 |
| maze2d-large | Diffuser[3] | 0.00 | 12.400 ± 2.440 | 0.0022 |
| | Conditional[4] | 194.31 ± 13.25 | 1.653 ± 1.124 | 0.0027 |
| | SafeDiffuser[9] | 196.25 ± 33.06 | **0.000 ± 0.000** | 0.2219 |
| | Projected[10] | 199.95 ± 37.45 | **0.000 ± 0.000** | 0.0083 |
| | Primal-Dual | 187.52 ± 33.80 | 0.069 ± 0.056 | **0.0028** |
| | ALM | 195.85 ± 21.82 | 0.033 ± 0.044 | 0.0029 |

## 7.2 Safe control in locomotion tasks

We evaluate our algorithm on planning and control in two Gymnasium MuJoCo environments, Hopper and Swimmer [51]. In Hopper, we bound the thigh-hinge angular velocity; in Swimmer, we bound the second rotor's angular velocity. We impose DCBF constraints and use an IDM to generate actions. Baselines are Diffuser, Conditional Diffuser, and SafeDiffuser. For the Projected Gradient Descent methods, we adapt the approach from [10] to handle sample-based constraints. However, compare to [10], for online execution, we further embed DCBF constraints rather than simple state-based hard constraints to smooth the resulting trajectories with a receding-horizon loop that executing only the first action of a planned sequence before replanning. This ensures the IDM can generate feasible and effective control solutions in real-time. Metrics include (i) rewards, (ii) planning constraint violations for 1000 steps, (iii) online violations and rates under IDM execution, and (iv) per-step diffusion compute time (Table 2). In Hopper, Projected achieves zero planning violations but lower online returns and occasional violations from IDM generalization and stochasticity, reduced by 90% vs. Diffuser. PD and ALM cut planning violations by 96% and yielding higher rewards and an 85% violation reduction, at just $1/400$ the computation time of [9, 10]. In Swimmer, Projected reduces online violations by only 50%, whereas PD/ALM achieve 98%, which we attribute to better alignment with the expert distribution and hence safer IDM actions. We also compare the constraint violations during both the planning and implementation stage with and without DCBF as a constraint (directly use hard constraints). The results demonstrate that both planned violations and implementing violations can be effectively reduced by incorporating DCBFs. Based on our analysis, we conclude that when constraints are directly applied, the IDM may fail to generate a reasonable and feasible action, validating that DCBF improves safety during online execution. In addition, we perform a sensitivity analysis of the IDM model to assess its impact on the control stage in Appendix F.

Table 2: Performance comparison for locomotion tasks. We report the mean and standard error over 10 random environments.

| Env | Algorithm | Reward | Planning Violations | Planning (w/o CBF) | Impl. Violations | Impl. (w/o CBF) | Violation Rate (%) | Time (s) |
|---|---|---|---|---|---|---|---|---|
| Hopper | Diffuser[3] | 3592 ± 37 | 2.55 ± 0.21 | - | 2.61 ± 0.21 | - | 4.82 ± 0.60 | 0.0026 |
| | Conditional[4] | 3608 ± 36 | 1.06 ± 0.19 | 1.21 ± 0.33 | 1.38 ± 0.27 | 1.14 ± 0.27 | 4.74 ± 0.83 | 0.0027 |
| | SafeDiffuser[9] | 3514 ± 46 | **0.00 ± 0.00** | **0.00 ± 0.00** | **0.28 ± 0.06** | 1.13 ± 0.22 | 0.27 ± 0.05 | 0.9307 |
| | Projected[10] | 3547 ± 13 | **0.00 ± 0.00** | **0.00 ± 0.00** | 0.29 ± 0.09 | 0.42 ± 0.10 | 0.29 ± 0.08 | 0.9305 |
| | Primal-Dual | 3551 ± 10 | 0.10 ± 0.03 | 0.13 ± 0.03 | 0.43 ± 0.12 | 0.46 ± 0.43 | 0.41 ± 0.30 | **0.0027** |
| | ALM | 3553 ± 14 | 0.08 ± 0.02 | 0.10 ± 0.03 | 0.38 ± 0.16 | 0.61 ± 0.12 | **0.09 ± 0.11** | 0.0031 |
| Swimmer | Diffuser[3] | 57.4 ± 8.7 | 8.38 ± 0.87 | - | 1.12 ± 0.26 | - | 1.30 ± 0.26 | 0.0026 |
| | Conditional[4] | 58.8 ± 5.6 | 8.02 ± 0.78 | 5.21 ± 0.66 | 1.00 ± 0.31 | 0.93 ± 0.44 | 1.11 ± 0.43 | 0.0027 |
| | SafeDiffuser[9] | 60.4 ± 6.1 | **0.00 ± 0.00** | **0.00 ± 0.00** | 0.65 ± 0.21 | 0.78 ± 0.38 | 0.97 ± 0.25 | 0.9301 |
| | Projected[10] | 58.7 ± 4.1 | **0.00 ± 0.00** | **0.00 ± 0.00** | 0.61 ± 0.25 | 0.75 ± 0.29 | 0.81 ± 0.26 | 0.9049 |
| | Primal-Dual | 86.4 ± 9.2 | 0.17 ± 0.08 | 0.16 ± 0.06 | 0.02 ± 0.02 | 0.06 ± 0.08 | 0.07 ± 0.07 | **0.0027** |
| | ALM | 88.7 ± 5.4 | 0.08 ± 0.05 | 1.80 ± 0.52 | **0.00 ± 0.02** | 0.42 ± 0.24 | **0.03 ± 0.04** | 0.0029 |

### 7.3  Adaptability to Time-varying Constraints

Finally, we evaluate the adaptability of our proposed algorithm in the PyBullet SafetyBallRunning environment, where safety constraints vary over time. In this setup, we have an obstacle ball that moves in the space, requiring the ball to avoid collision while continuing to move. We assume that the agent ball has no knowledge of the obstacle's dynamics and can only observe its current states, making planning infeasible. During implementation, we construct the discrete control barrier functions (DCBF) based on the current observation of obstacles as constraints in the constrained diffuser to generate a trajectory. Following the same receding-horizon strategy and compared to [10], this approach ensures the IDM produces feasible actions. We use Diffuser and Conditional Diffuser as baseline algorithms. We compare rewards, constraint violations and violation rates during implementation stage, and computation time per diffusion step in the experiment. The results are summarized in Table 3. Ablation studies on hyperparameter of DCBF and worst-case analysis of initial conditions can be found in Appendices E and G.

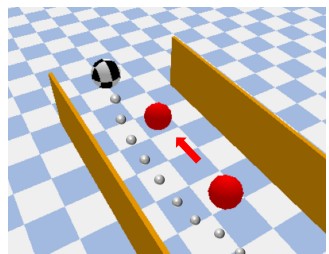

Figure 5: PyBulletBallRunning tasks with the black-white ball and a red moving obstacles ball.

Table 3: Performance comparison for PyBullet ball running tasks. The results indicate our methods effectively adapt to time-varying constraints. We report the mean and the standard error over 10 random environments.

| Env | Algorithm | Reward | Impl. Violations | Impl. (w/o CBF) | Violation Rate (%) | Time (s) |
|---|---|---|---|---|---|---|
| SafetyBall Running | Diffuser[3] | $646.4 \pm 30.3$ | $2.20 \pm 0.77$ | - | $37.7 \pm 5.1$ | 9.6e-3 |
| | Conditional[4] | $616.5 \pm 33.9$ | $1.55 \pm 0.53$ | $1.99 \pm 1.09$ | $33.1 \pm 11.1$ | 9.7e-3 |
| | SafeDiffuser[9] | $820.8 \pm 29.8$ | $1.13 \pm 1.08$ | $1.36 \pm 0.46$ | $21.9 \pm 12.8$ | 2.6e-2 |
| | Projected[10] | $586.7 \pm 33.7$ | $1.01 \pm 0.63$ | $1.50 \pm 0.14$ | $25.8 \pm 13.9$ | 2.6e-2 |
| | Primal-Dual | $722.2 \pm 118.6$ | $0.02 \pm 0.03$ | $0.10 \pm 0.13$ | $2.2 \pm 3.2$ | 1.3e-2 |
| | ALM | $696.1 \pm 99.9$ | $\mathbf{0.02 \pm 0.02}$ | $\mathbf{0.02 \pm 0.02}$ | $\mathbf{1.6 \pm 3.4}$ | 2.1e-2 |

Compared to the baselines, we can ensure effective collision reduction with less decision time. The Projected method reduces violations by only 50%, with 170% more computation time based on the closed-form projection. In contrast, PD and ALM achieve 99% reduction, with 35% and 119% computation overhead, respectively. In addition, we also observed an improvement in the reward during implementation. We interpret this as the adjustment bringing the ball closer to the target.

## 8  Conclusions

In this work, we proposed constrained diffusers for planning and control tasks. The model integrates constraints over distributions directly into pre-trained diffusion models without requiring retraining or architectural changes. By re-interpreting safe trajectory generation as a constrained sampling problem, we apply three methods–Projected, Primal-Dual and Augmented Lagrangian methods–to realize the constrained reverse Langevin sampling. Then we introduce DCBFs as constraints in the Constrained Diffusers and use inverse dynamics models to obtain actions, following a receding-horizon control that only the first action of the selected plan is executed before replanning, ensuring safety in the implementation stage. We demonstrated them in Maze2D, robotic locomotion, and PyBullet ball running tasks showing that the proposed constrained diffusion model not only meets safety requirements but also reduces computational time compared to existing approaches. We summarize the limitations into theoretical and practical ones. On the theoretical end, our methods rely on assumptions including accurate score function estimation. On the practical end, we made an assumption that we have access to constraints, and lack robustness analysis. For a detailed analysis, please refer to Appendix B.

## Acknowledgments and Disclosure of Funding

AP was supported in part by UK's Engineering and Physical Sciences Research Council projects EP/X017982/1, EP/Y014073/1 and UKRI2108. JU was supported by the University of Oxford's Strategic Research Fund through the University of Oxford's ZERO Institute.

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

# A  Comparisons on Related Works

Compared with Projected Diffusion [10], which enforces feasibility by projecting each sampled point onto a predefined safe set, our Constrained Diffusers treat generation as constrained sampling at the distribution level. Beyond the projected method, we introduce primal–dual and augmented Lagrangian updates that support distribution-level constraints, jointly update trajectories and Lagrange multipliers inside the reverse process. For online execution, we further embed DCBF constraints rather than state-based hard constraints and use a receding-horizon loop that executes only the first action before replanning, avoiding per-step QP solves and aligning the generated trajectory with system dynamics. Under some assumptions we establish convergence and constraint-satisfaction guarantees, and empirically show in Maze2D, locomotion, and PyBullet tasks that our methods satisfy constraints with less computation while matching or improving task performance.

We provide a comparative analysis of several key approaches in safety-critical planning and control in Table 4.

Table 4: Performance Comparison

| Method | Constraint Satisfaction | Computational Efficiency | Primary Application / Notes |
|---|---|---|---|
| Traditional QP with CBF | Strict; can find an optimal safe plan with minimal costs if one exists. | Average; solving a QP for each action (e.g., via primal-dual interior point method). | Known dynamics and cost functions. |
| SafeDiffuser | High; Planning: solves a QP at each diffusion denoising step to ensure the planned trajectory is safe. Inference: uses state-based hard constraints. | Low; solving a QP at every step of the diffusion process is computationally expensive. | Safety-critical systems where having a guaranteed safe plan is essential. |
| Projected Diffusion (Ours and [10]) | Strict; Planning: guarantees the planned trajectory has zero violations. Inference: DCBF as constraints, violations can occur due to errors from the IDM or environment randomness. | Low, solving an optimization problem each diffusion denoising step. | Systems that require strict safety guarantees during the planning phase. |
| Primal-Dual Diffusion (Ours) | Good; Planning: may have very small violations in the final plan due to the iterative process. Inference: keeps the system safe with DCBF constraints, especially when the planned path is similar to expert behavior. | Extremely high; uses simple gradient updates and avoids solving optimization problems. | Real-time applications where planning speed is more important than perfect constraint satisfaction. |
| ALM Diffusion (Ours) | Strong; Planning: produces plans with near-zero violations; Inference: provides robust online safety with DCBF. | High; slightly slower than the PD method but much faster than solving QPs. | General use cases that need a good balance between safety and planning speed. |

# B  Limitations

We categorize limitations into theoretical and practical aspects.

**Theoretical Aspects.**  Our theoretical derivations and analyses rely on assumptions that may not hold perfectly in practice.

- **Score Estimation Accuracy:** We assume that the trained neural network yields an accurate estimate of the score function ($\nabla_{\mathbf{x}_t} \log p_t(\mathbf{x}_t)$). Our analysis does not quantify how these errors impact the probability or degree of constraint violation in the final samples.
- **Convergence Guarantees:** Our convergence analysis relies on standard, but potentially strong, assumptions regarding the score and constraint functions. Furthermore, the results established provide guarantees for local, rather than global, convergence.

**Practical Aspects.**    From a practical standpoint, the applicability and scope of the current method can be extended.

- **Constraint Complexity and Guarantees:** We have not addressed more complex scenarios involving, e.g., constraints coupled with system dynamics or other constraints beyond safety. Moreover, while we empirically demonstrate a high rate of satisfaction, the proposed method does not, in its current form, offer formal, provable guarantees of 100% constraint satisfaction for every sample, which may be a requirement for safety-critical applications.
- **Known and Differentiable Constraints:** A core assumption is that the constraints $g(x)$ are known in an explicit, analytical form, and that their gradients $\nabla g(x)$ are readily available.
- **Robustness to Uncertainty:** The formulation assumes that constraints are deterministic. We do not currently address how to handle uncertainty in the environment, which is vital for robust deployment in noisy or variable conditions.

## C  Experimental Details

The experiments are all running on the machine with Intel Core i7-14700 $\times$ 28, 32GB Memory, NVIDIA GeForce RTX 4060 and the operational system Ubuntu 24.04. All diffuser models in our experiments were trained from scratch using expert demonstration data. For Maze2D, we used Maze2D-umaze-v1 and Maze2D-large-v1 from D4RL; for MuJoCo locomotion tasks, we used hopper-medium-expert-v2 from D4RL and swimmer-medium-expert-v2 from Minari; for PyBullet ball running tasks, we used SafetyBallRun-v0 from DSRL.

The experimental details are summarized as follows:

- In Maze2D environment, the noise is approximated by a temporal U-Net [3]. The training settings and codes are borrowed from [3, 9]. For Maze2D-umaze tasks, we use a planning horizon of 128, 64 diffusion steps and additional 64 steps in reverse process. The constraint is formulated as $x + y \leq b$, where $(x, y)$ is the position state of the agent and $b$ is the parameter of the boundary. The learning rate $\eta$ for dual variables in primal-dual methods is 0.025 and the initial penalty $\rho$ is 0.05. For Maze2D-large tasks, we use a planning horizon of 384, 256 diffusion steps and additional 200 steps in reverse process. The constraint is formulated as $(\frac{x-x_0}{a})^2 + (\frac{y-y_0}{b})^2 \geq 1$, where $(x, y)$ is the position state of the agent and $(x_0, y_0, a, b)$ are the parameters of the boundary. The learning rate $\eta$ for dual variables in primal-dual methods is $1e^{-3}$ and the initial penalty $\rho$ is $2.5e^{-4}$.
- In Gymnasium Mujoco and PyBullet environment, the temporal U-Net architecture, training settings and codes are borrowed from [4]. For Hopper tasks, we use a planning horizon of 100, 200 diffusion steps and additional 100 steps in reverse process. The constraint is formulated as $\omega \leq \omega_{\max}$, where $\omega$ is the angular velocity of the thigh hinge. The learning rate $\eta$ for dual variables in primal-dual methods is 0.1 and the initial penalty $\rho$ is 0.01. The hyperparameter $\alpha$ in DCBFs is 0.85. For Swimmer, we use a planning horizon of 100, 200 diffusion steps and additional 100 steps in reverse process. The constraint is formulated as $\omega \leq \omega_{\max}$, where $\omega$ is the angular velocity of the second rotor. The learning rate $\eta$ for dual variables in primal-dual methods is 0.5 and the initial penalty $\rho$ is 0.01. The hyperparameter $\alpha$ in DCBFs is 0.85. For SafetyBallRunning tasks, we use a planning horizon of 100, 200 diffusion steps and additional 100 steps in reverse process. The constraint is formulated as $[x - (x_0 + v_x\tau)]^2 + [y - (y_0 + v_y\tau)]^2 \geq (R + r)^2$, where $(x, y)$ is the position state of the ball agent, $(x_0, y_0)$ represents the position of the obstacle ball, $(v_x, v_y)$ represents the velocity, $R$ represents the radius of agent ball and $r$ represents the radius of the obstacle ball. The learning rate $\eta$ for dual variables in primal-dual methods is 0.1 and the initial penalty $\rho$ is 0.01. The hyperparameter $\alpha$ in DCBFs ranges from 0.3 to 0.8.

We assume access to a differentiable constraint function and its gradient: for common geometric obstacles these are analytic; in more complex settings, constraints can be built online from sensor data (e.g., camera-based segmentation of obstacle boundaries). For black-box constraints, a neural network learns a differentiable approximation from cost signals collected via interaction with the environment.

## D  Scalability to high-dimensional action space in manipulation tasks

To demonstrate the wider applicability of the algorithm, we conducted additional experiments on the Dexterous Hand Manipulation Adroit Relocate task with a 30-dimensional action space. The new environment Adroit Relocate was introduced in [52] as part of the Adroit manipulation platform. For our new experiments, we use the D4RL dataset relocate-expert-v1, which contains 5,000 expert trajectories sampled from an expert that solves the task. We imposed two constraints: one is that the translation distance of the full arm at each step cannot exceed a certain value ($x^2 + y^2 + z^2 \leq r^2$), and the other is that the angular up-and-down movement of the full arm cannot exceed a certain value. Note that since the robot's action is its end-effector pose, the constraints are directly applied to the action. Therefore, it is not necessary to use an IDM and compare planning violations in this task. Results summarized below show that even with similarly sized models, our algorithms satisfy constraints, demonstrating scalability to high-dimensional tasks.

Table 5: Performance comparison for relocate tasks. We can find that our algorithm could almost satisfy constraints and reduce computational time, compared to baseline algorithms. We report the mean and the standard error over 10 random environments.

| Env | Algorithm | Rewards | Violations | Violation Rate (%) | Time (s) |
|---|---|---|---|---|---|
| Relocate | Diffuser[3] | 550±938 | 87.0±30.4 | 74.0±16.4 | 0.0028 |
| | Conditional[4] | 422±837 | 32.0±14.7 | 58.5±16.3 | 0.0029 |
| | Projected[10] | 321±791 | 0.0±0.0 | 0.0±0.0 | 0.9515 |
| | Primal-Dual | 499±951 | 0.04±0.03 | 2.9±1.9 | 0.0031 |
| | ALM | 325±670 | 0.04±0.02 | 2.9±1.6 | 0.0035 |

## E  Ablation Studies on hyperparameters

In the control tasks, we apply discrete control barrier functions as constraints in the reverse diffusion process. We find that $\alpha$ plays an important role in the safety implementation. To show its importance, we compare the performance of our Constrained Diffusers with different values of $\alpha$ in SafetyBall-Running tasks. Also, when we use hard constraints rather than DCBFs, it is equivalent to $\alpha = 1$. Theoretically, smaller $\alpha$ may lead to a more robust solution which contradicts the final results. This may occur as a result of the smaller $\alpha$ making the trajectories deviate more from the distribution and the inverse dynamics model is not accurate enough. Results are summarized in Table 6

Table 6: Performance comparison for PyBullet SafetyBallRunning tasks with different $\alpha$. We report the mean and the standard error over 10 random environments.

| $\alpha$ | Algorithm | Reward | Total Impl. Violations | Violation Rate (%) |
|---|---|---|---|---|
| $\alpha = 0.3$ | Projected[10] | 586.7 ± 33.7 | 1.01 ± 0.63 | 25.8 ± 13.9 |
| | Primal-Dual | 717.3 ± 107.7 | 0.21 ± 0.23 | 10.4 ± 11.38 |
| | ALM | 696.0 ± 144.5 | 0.29 ± 0.19 | 9.3 ± 9.87 |
| $\alpha = 0.5$ | Projected[10] | 595.3 ± 46.3 | 1.47 ± 0.50 | 29.7 ± 10.5 |
| | Primal-Dual | 720.6 ± 80.97 | 0.09 ± 0.09 | 9.2 ± 7.9 |
| | ALM | 647.7 ± 74.34 | 0.17 ± 0.17 | 10.3 ± 9.8 |
| $\alpha = 0.8$ | Projected[10] | 595.7 ± 19.6 | 1.54 ± 0.53 | 36.0 ± 1.41 |
| | Primal-Dual | 722.2 ± 118.6 | 0.02 ± 0.03 | 2.2 ± 3.2 |
| | ALM | 696.1 ± 99.9 | **0.02 ± 0.02** | **1.6 ± 3.4** |

# F  Sensitivity Analysis to IDM Prediction Errors

The accuracy of the IDM is critical because errors cause a mismatch between planned and actual states, leading to potential task failures and safety violations. We quantify this sensitivity by defining the IDM error as $\Delta u \approx \tilde{u} - u^*$. Linearizing the dynamics shows the resulting state deviation $\Delta x \approx \frac{\partial f}{\partial u} \Delta u$, which affects the safety function as $h(x_{\text{actual}}^{t+1}) \approx h(x_{\text{plan}}^{t+1}) + \nabla h(x_{\text{plan}}^{t+1})^\top \Delta x$, meaning the change in constraints is primarily influenced by $\nabla h(x_{\text{plan}}^{t+1})^\top \frac{\partial f}{\partial u}$. To validate the capability of our method to maintain safety constraints in the presence of prediction errors, we conducted a sensitivity analysis on Hopper. We evaluated three algorithms (Projected, Primal-Dual, ALM) with varying levels of IDM errors (2%, 10%, 25%). Results show that increasing IDM error generally lowers reward and increases constraint violations, but a 10% error is still acceptable for the tasks. The results are shown in Table 7.

Table 7: Sensitivity analysis on IDM errors in Hopper and Swimmer

| Env | Error | Algorithm | Reward | Impl. Violations | Violation Rate (%) |
|---|---|---|---|---|---|
| Hopper | 2% | Projected[10] | 3437±196 | 0.28±0.12 | 1.76±0.75 |
| | | Primal-Dual | 3501±130 | 0.48±0.21 | 3.47±0.85 |
| | | ALM | 3370±442 | 0.44±0.19 | 3.13±0.10 |
| | 10% | Projected[10] | 3391±325 | 0.32±0.08 | 1.98±0.60 |
| | | Primal-Dual | 2690±707 | 0.46±0.12 | 3.33±0.86 |
| | | ALM | 2369±455 | 0.41±0.08 | 3.51±0.02 |
| | 25% | Projected[10] | 1900±471 | 0.62±0.12 | 2.09±0.29 |
| | | Primal-Dual | 1446±354 | 1.19±0.22 | 4.67±0.75 |
| | | ALM | 1614±504 | 0.95±0.33 | 4.11±0.02 |
| Swimmer | 2% | Projected[10] | 58.1±4.7 | 0.58±0.31 | 0.72±0.31 |
| | | Primal-Dual | 89.4±6.2 | 0.05±0.06 | 0.10±0.07 |
| | | ALM | 89.2±5.5 | 0.03±0.04 | 0.05±0.08 |
| | 10% | Projected[10] | 62.1±5.8 | 0.66±0.23 | 0.77±0.28 |
| | | Primal-Dual | 87.6±7.3 | 0.03±0.08 | 0.09±0.08 |
| | | ALM | 88.9±5.3 | 0.03±0.06 | 0.07±0.10 |
| | 25% | Projected[10] | 61.5±5.2 | 0.73±0.31 | 0.69±0.29 |
| | | Primal-Dual | 86.3±5.9 | 0.07±0.03 | 0.50±0.07 |
| | | ALM | 92.8±4.5 | 0.04±0.30 | 0.30±0.45 |

# G  Worst-Case Analysis under Extreme Initial Conditions

To show the robustness of our methods under extreme initial conditions, we performed additional worst-case analysis in the PyBullet Safety BallRunning environment. In the original setup, the agent and the moving obstacle start with a safe distance between them. In this new analysis, we created more extreme initial conditions by placing the obstacle closer to the agent, reducing the initial safety distance. The results are summarized in Table 8. The "SafetyMargin" denotes the starting distance as a percentage of the sum of the radii of the ball and the obstacle, with "Boundary" meaning the agent and obstacle start at the point of collision.

Even in the most extreme case starting at the point of collision, our methods still effectively reduce the magnitude and frequency of constraint violations compared to the baseline. As the initial safety margin increases to 10% and 25%, the number of violations is further and substantially reduced, showcasing the framework's ability to maintain safety even when initialized in or very near to a compromised state.

Table 8: Worst-Case initial case analysis in Safety BallRunning task

| SafetyMargin | Algorithm | Reward | Impl.Violations | Violation Rate (%) |
|---|---|---|---|---|
| Boundary | Diffuser[3] | 640.6±29.0 | 14.04±1.27 | 81.9±1.92 |
| | Projected[10] | 536.7±36.2 | 3.44±0.32 | 74.6±1.6 |
| | Primal-Dual | 457.1±48.4 | 2.99±0.12 | 67.5±2.3 |
| | ALM | 380.9±27.1 | 3.10±0.17 | 70.1±1.5 |
| 10% | Diffuser[3] | 640.8±28.9 | 13.08±1.40 | 76.5±2.2 |
| | Projected[10] | 507.1±57.6 | 2.63±0.36 | 57.8±2.0 |
| | Primal-Dual | 479.5±54.6 | 1.55±0.23 | 32.5±5.6 |
| | ALM | 421.9±59.77 | 1.33±0.21 | 29.0±4.4 |
| 25% | Diffuser[3] | 640.6±29.0 | 11.42±1.65 | 69.2±2.99 |
| | Projected[10] | 525.0±59.2 | 0.31±0.16 | 19.6±3.7 |
| | Primal-Dual | 413.9±87.9 | 0.45±0.75 | 18.9±16.5 |
| | ALM | 422.1±70.4 | 0.06±0.11 | 8.0±8.5 |

## H   Variations of executing steps

We conducted an experiment to evaluate the impact of executing multiple steps (1, 2, and 3) before replanning in the Swimmer task. The results are summarized in Table 9.

The results demonstrate that performance degrades dramatically for all methods when executing two or more steps at a time before replanning. This finding supports our hypothesis that multi-step execution strategies are highly sensitive to the accumulation of errors, which can stem from the generalization of the diffusion model and potential inaccuracies in the inverse dynamics model. Therefore, a single-step, "closed-loop" execution strategy, where replanning occurs at each timestep, proves to be the most robust and effective approach for achieving high performance in our experiments.

Table 9: Performance and constraint violations under different numbers of execution steps before replanning in the Swimmer task. We report the mean and standard error over 10 random seeds.

| Steps | Algorithm | Reward | Impl. Violations | Violation Rate (%) |
|---|---|---|---|---|
| 1 step | Diffuser[3] | 57.4 ± 8.7 | 1.12 ± 0.26 | 1.30 ± 0.26 |
| | Projected[10] | 58.7 ± 4.1 | 0.61 ± 0.25 | 0.81 ± 0.26 |
| | Primal-Dual | 86.4 ± 9.2 | 0.02 ± 0.02 | 0.07 ± 0.07 |
| | ALM | 88.7 ± 5.4 | 0.00 ± 0.02 | 0.03 ± 0.04 |
| 2 steps | Diffuser[3] | 12.4 ± 25.9 | 0.09 ± 0.15 | 0.20 ± 0.24 |
| | Projected[10] | 8.2 ± 13.2 | 0.00 ± 0.00 | 0.00 ± 0.00 |
| | Primal-Dual | 7.9 ± 25.5 | 0.05 ± 0.08 | 0.13 ± 0.20 |
| | ALM | 2.4 ± 12.3 | 0.00 ± 0.01 | 0.01 ± 0.03 |
| 3 steps | Diffuser[3] | 0.7 ± 16.7 | 0.04 ± 0.12 | 0.06 ± 0.15 |
| | Projected[10] | 5.2 ± 20.1 | 0.02 ± 0.06 | 0.04 ± 0.09 |
| | Primal-Dual | 5.9 ± 13.7 | 0.01 ± 0.03 | 0.04 ± 0.08 |
| | ALM | 8.9 ± 23.2 | 0.00 ± 0.00 | 0.00 ± 0.00 |

## I   Analysis of Projected Methods

Recall that the projected Langevin sampling can be written as

$$x_{t-1} = \Pi_{\mathcal{C}}\left(x_t + \frac{\beta_t}{2}\nabla_{x_t}\log p(x_t) + \sqrt{\beta_t}z\right). \tag{20}$$

From [15], we could conclude that when the distribution is log-concave with smooth potential and there exists a convex body $K \subset \mathbb{R}^n$ that $0 \in K$, $K$ contains a Euclidean ball of radius $r$, and $K$ is

contained in a Euclidean ball of radius $R$, projected Langevin Monte Carlo will target at sampling from distribution $q \propto e^{-f(x)\mathbb{I}\{x \in K\}}$. While in our problem settings, the distributions are not always log-concave and the constrained sets are not always convex. Therefore, it is always difficult to analyze it from the perspective of distributions. In the following part, we prove that the samples from projected methods will converge to a nearly stationary point under certain assumptions.

**Assumption 1.** *We assume that the iterates satisfy:*

1. $\log p(x)$ *is bounded and has L-Lipschitz continuous gradient:*

$$\|\nabla \log p(x) - \nabla \log p(y)\| \le L\|x - y\| \quad \forall x, y \in \mathcal{C} \tag{21}$$

2. $\sum_{t=0}^{\infty} \beta_t = \infty, \sum_{t=0}^{\infty} \beta_t^2 < \infty$

*Remark* 2. Since we interpret the reverse diffusion process as a Langevin sampling process, in our theoretical analysis, the iteration flows from $t = 0$ to $t = T$ when $T \to \infty$, rather than following the backward time direction in the original reverse diffusion process.

**Theorem I.1.** *Under the above assumptions, when $\beta_t \le \frac{1}{L}$ for all $t$ and we define the gradient mapping as $G(x_t) = \frac{2}{\beta_t}(x_t - x_{t-1})$, then when $t \to \infty$ samples from the projected methods satisfies:*

$$\inf \mathbb{E}[\|G(x_t)\|^2] \to 0 \tag{22}$$

*Proof.* Because of the Lipschitz gradient assumption, we have:

$$\log p(x_{t-1}) \le \log p(x_t) + \langle \nabla \log p(x_t), x_{t-1} - x_t \rangle + \frac{L}{2}\|x_{t-1} - x_t\|^2 \tag{23}$$

From the projection property $\langle y - \Pi_{\mathcal{C}}(y), x - \Pi_{\mathcal{C}}(x) \rangle \le 0$, we have:

$$\langle x_{t-1} - x_t - \frac{\beta_t}{2}\nabla \log p(x) - \sqrt{\beta_t}z, x_t - x_{t-1} \rangle \ge 0 \tag{24}$$

$$\langle x_t - x_{t-1}, \frac{\beta_t}{2}\nabla \log p(x) + \sqrt{\beta_t}z \rangle \ge \|x_{t-1} - x_t\|^2 \tag{25}$$

Hence, we can conclude that

$$\langle \nabla \log p(x_t), x_{t-1} - x_t \rangle \le -\frac{2}{\beta_t}\langle x_t - x_{t-1}, \frac{2}{\sqrt{\beta_t}}z \rangle - \frac{2}{\beta_t}\|x_{t-1} - x_t\|^2 \tag{26}$$

Then we combine (23) and (26),

$$\log p(x_{t-1}) \le \log p(x_t) - \frac{2}{\beta_t}\langle x_t - x_{t-1}, \frac{2}{\sqrt{\beta_t}}z \rangle - \frac{2}{\beta_t}\|x_{t-1} - x_t\|^2 + \frac{L}{2}\|x_{t-1} - x_t\|^2 \tag{27}$$

$$= \log p(x_t) - \left(\frac{2}{\beta_t} - \frac{L}{2}\right)\|x_{t-1} - x_t\|^2 - \frac{2}{\beta_t}\langle x_t - x_{t-1}, \frac{2}{\sqrt{\beta_t}}z \rangle \tag{28}$$

Then we take expectations on the equations,

$$\mathbb{E}[\log p(x_{t-1})] \le \mathbb{E}[\log p(x_t)] - \left(\frac{2}{\beta_t} - \frac{L}{2}\right)\mathbb{E}[\|x_{t-1} - x_t\|^2] \tag{29}$$

since $\mathbb{E}[\langle x_t - x_{t-1}, \frac{2}{\sqrt{\beta_t}}z \rangle] = 0$. Then we define $\delta_t := \left(\frac{2}{\beta_t} - \frac{L}{2}\right)$ and sum over $t = T, \dots, 0$

$$\sum_{t=0}^{T} \frac{1}{\delta_t}\mathbb{E}[\|x_{t-1} - x_t\|^2] \le \log p(x_0) - \log p(x_T) \tag{30}$$

Since $\log p(x)$ is bounded, we can rewrite (30) as

$$\sum_{t=0}^{T} \frac{1}{\delta_t}\mathbb{E}[\|x_{t-1} - x_t\|^2] \le \log p(x_0) - \inf \log p(x) < \infty \tag{31}$$

Then we relate (31) to the projected methods. Note that the update rule of projected methods is $x_{t-1} = \Pi_{\mathcal{C}} \left( x_t + \frac{\beta_t}{2} \nabla_{x_t} \log p(x_t) + \sqrt{\beta_t} z \right)$. Recall the definition of gradient mapping $G(x_t) = \frac{2}{\beta_t}(x_t - x_{t-1})$. Then, we have the following inequality according to (31),

$$\sum_{t=0}^{T} \beta_t^2 \mathbb{E}[||G(x_t)||^2] < \infty \tag{32}$$

Since $\sum_{t=0}^{T} \beta_t = \infty$, we have

$$\inf \mathbb{E}[||G(x_t)||^2] \to 0 \tag{33}$$

when $t$ goes to $\infty$. $\qquad \square$

Under the given conditions, as the number of iterations $k$ tends to infinity, the inferior of the expected squared norm of the gradient mapping $G(x_t)$ goes to zero. In other words, the algorithm approaches a stationary point during the iterations.

## J   Proof of Primal-Dual Methods

We employ the multi-timescale stochastic approximation framework [18] to analyze the convergence of the primal-dual update and show that the convergent results satisfy the constraints. Combined with Remark 2, we make the following assumptions that the step sizes satisfy the standard multi-timescale approximation framework conditions:

**Assumption 2** (Step Sizes). *The step size schedules $\{\beta_t\}$ and $\{\eta_t\}$ satisfy:*

1. *$\sum_{t=0}^{\infty} \beta_t = \infty, \quad \sum_{t=0}^{\infty} \eta_t = \infty$ .*

2. *$\sum_{t=0}^{\infty} \beta_t^2 < \infty, \quad \sum_{t=0}^{\infty} \eta_t^2 < \infty$*

3. *$\eta_t = o(\beta_t)$*

The condition $\eta_t = \mathcal{O}(\beta_t)$ ensures that the primal variables $(x_t)$ evolve on a faster timescale than the dual variables $(\lambda_t)$.

**Step 1 (Convergence of primal update)** Consider the dynamics of $x_t$ while holding the dual variable $\lambda_t$ fixed at some value $\bar{\lambda}$. The primal update (13) becomes:

$$x_{t+1} = x_t + \frac{\beta_t}{2}(\nabla_{x_t} \log p(x_t) - \bar{\lambda}^{\mathsf{T}} \nabla_{x_t} g(x_t)) + \sqrt{\beta_t} z_t. \tag{34}$$

Defining the potential function of the distribution as $U(x_t, \bar{\lambda}) = -\log p(x_t) + \bar{\lambda}^{\mathsf{T}} g(x_t)$, we have that $\nabla U(x_t, \bar{\lambda}) = -\nabla_{x_t} \log p(x_t) + \bar{\lambda}^{\mathsf{T}} \nabla_{x_t} g(x_t)$. Consequently the update can be written as:

$$x_{t+1} = x_t - \frac{\beta_t}{2} \nabla U(x_t, \bar{\lambda}) + \sqrt{\beta_t} z_t. \tag{35}$$

This is the Euler-Maruyama discretization of the overdamped Langevin stochastic differential equation (SDE). The update rule (34) falls under the Langevin-Robbins-Monro (LRM) scheme described in Theorem 4.9, Section 4.4 of [14]. The theorem states that for a Langevin-Robbins-Monro (LRM) scheme of the form:

$$x_{n+1} = x_n + \gamma_{n+1}\{v(x_n) + Z_{n+1}\} + \sqrt{\gamma_{n+1}} \sigma(x_n) \xi_{n+1} \tag{36}$$

where $\xi_{n+1}$ are i.i.d. standard Gaussian, the stochastic interpolation $(X_t)_{t \geq 0}$ of $\{x_n\}_{n \in \mathbb{N}}$ is an asymptotic pseudo trajectory of the flow $\Phi$ corresponding to the SDE

$$dY_t = v(Y_t)dt + \sigma(Y_t)dW_t \tag{37}$$

in the Wasserstein space, provided the some assumptions for theorem 4.9 in [14] hold. We now verify the assumptions for our specific update.

**A1 (Lipschitz):** Firstly, we need the drift term $v(x) = \nabla \log p(x) - \bar{\lambda}^{\mathsf{T}} \nabla g(x)$ to be Lipschitz. We make the following assumption for the proof:

**Assumption 3.** *Assume that the gradients $\nabla \log p(x)$ and $\nabla g(x)$ are Lipschitz continuous with constants $L_p$ and $L_g$, respectively.*

Under Assumption 3, $v(x)$ is Lipschitz continuous:

$$
\begin{aligned}
\|v(x) - v(y)\| &= \left\| (\nabla \log p(x) - \bar{\lambda}^T \nabla g(x)) - (\nabla \log p(y) - \bar{\lambda}^T \nabla g(y)) \right\| \\
&\leq \|\nabla \log p(x) - \nabla \log p(y)\| + \|\bar{\lambda}^T (\nabla g(x) - \nabla g(y))\| \\
&\leq L_p \|x - y\| + \|\bar{\lambda}\| \|\nabla g(x) - \nabla g(y)\| \\
&\leq L_p \|x - y\| + \|\bar{\lambda}\| L_g \|x - y\| \\
&= (L_p + \|\bar{\lambda}\| L_g) \|x - y\|
\end{aligned}
$$

Therefore, $v(x)$ is Lipschitz with constant $L_v = (L_p + \|\bar{\lambda}\| L_g)$. For diffusion term, we have $\sigma(x) = I_d$ which is a constant function. Thus, $\sigma(x)$ is Lipschitz with constant $L_\sigma = 0$. In conclusion, under Assumption 3, Assumption 4.2 in [14] holds.

**A2 (Robbin-Monro):** Based on Assumption 2, we have $\sum_{t=0}^{\infty} \beta_t = \infty$ and $\sum_{t=0}^{\infty} \beta_t^2 < \infty$ which satisfies conditions in Assumption 4.3 in[14].

**A3 (Error term):** We have $Z_{n+1} = 0$, so $U_{n+1} = 0$ and $B_{n+1} = 0$. For Assumption 4.1 in [14], measurability holds for $Z_{n+1} = 0$. For Condition 4.14 in [14], $\mathbb{E}[U_{n+1}|\mathcal{F}_{\tau_{n-1}}] = 0$. $\sup_{n \in \mathbb{N}} \mathbb{E}\|U_{n+1}\|^2 = 0 < \infty$. For Condition 4.15, $\mathbb{E}[\|B_{n+1}\|^2|\mathcal{F}_n] = 0$. We need to check if $0 \lesssim \beta_n^2 \|v(x_n)\|^2 + \beta_n$. Since $\beta_n > 0$ and $\|v(x_n)\|^2 \geq 0$, this inequality holds trivially. Thus, Assumption 4.4 in [14] holds.

**A5 (Bounded Moments):** We need to verify that $\sup_{n \in \mathbb{N}} \mathbb{E}\|x_n\|^2 < \infty$. Theorem 4.13 in [14] provides sufficient conditions for this, requiring $v$ to be dissipative and either $L_\sigma^2 < \alpha$ (where $\alpha$ is the dissipativity constant) or $\sigma$ to be bounded.

**Definition J.1** (Dissipativity (Def 4.12 in [14]))**.** The drift $v$ is $(\alpha, \beta)$-dissipative if $\langle x, v(x) \rangle \leq -\alpha \|x\|^2 + \beta$ for some $\alpha > 0, \beta \geq 0$.

**Assumption 4.** *Assume the drift $v(x) = -\frac{1}{2}(\nabla \log p(x) - \bar{\lambda}^T \nabla g(x))$ is $(\alpha, \beta)$-dissipative for some $\alpha > 0, \beta \geq 0$.*

Also, we could find that $\sigma(x) = I_d$ is Lipschitz, $\lim_{n \to \infty} \beta_n = 0$, Condition (ii) of Theorem 4.13 in [14]requires $\sigma$ to be bounded in Frobenius norm. $\|\sigma(x)\|_F = \|I_d\|_F = \sqrt{d} < \infty$ all hold. Therefore, by Theorem 4.13, we have $\sup_{n \in \mathbb{N}} \mathbb{E}\|x_n\|^2 < \infty$.

At this point, all assumptions are satisfied and we could conclude that the update rule (2) is an asymptotic pseudo trajectory of the flow corresponding to the following SDE:

$$
dX_t = -\frac{1}{2} \nabla U(X_t, \bar{\lambda}) dt + dW_t. \tag{38}
$$

when $t \to \infty$, it converges to the stationary distribution of this SDE, denoted by $q_{\bar{\lambda}}$:

$$
q_{\bar{\lambda}}(x) \propto \exp(-\log p(x) + \lambda^\mathsf{T} g(x)). \tag{39}
$$

Furthermore, similar to Lemma 4.10 in [14], the relative entropy $H(\cdot|q_{\bar{\lambda}})$ acts as a Lyapunov function, implying that the only internally chain-transitive set for the SDE flow is the singleton $\{q_{\bar{\lambda}}\}$. This implies:

**Lemma J.1** (Fast Scale Convergence)**.** *Under the assumptions above, for any fixed $\bar{\lambda} \geq 0$, the distribution of $x_t$ generated by (34) converges to the unique stationary distribution $q_{\bar{\lambda}}$.*

**Step 2 (Convergence of dual update)** We analyze the slower dynamics of $\lambda_t$ based on the previous stationary distribution conclusion. However, in practice, the approximation of $\mathbb{E}_q[g(x)]$ may be inefficient. Therefore, we directly use the term $g(x)$ and rewrite the dual update:

$$
\lambda_{t+1} = [\lambda_t + \eta_t g(x_{t+1})]_+ \tag{40}
$$

Consider the associated ordinary differential equation (ODE) related to the behavior of $\lambda$ under the distribution $q_{\lambda_t}$:

$$
\frac{d\lambda_t}{dt} = [\nabla_\lambda L(q^*, \lambda)]_+ = [\mathbb{E}_{x \sim q_\lambda}[g(x)]]_+ \tag{41}
$$

where $t$ represents the slow time scale associated with $\eta_t$. Then, we prove that the update (40) is a stochastic approximation of the ODE (41) based on stochastic approximation theory. To illustrate the proof, we write the dual update into the following formulation:

$$\lambda_{t+1} = [\lambda_t + \eta_t(\nabla_\lambda L(q^*, \lambda) + M_{t+1})]_+, \tag{42}$$

where $M_{t+1}$ is Martingale differences w.r.t the $\sigma$-field $\sigma(\lambda_i, M_i, i < t)$ and here $M_{t+1} = g(x) - \nabla_\lambda L(q^*, \lambda) = g(x) - \mathbb{E}_{x \sim q_\lambda}[g(x)]$. Because the samples are generated independently and randomly from the distribution $q_{\lambda_t}$, we can easily obtain that $\mathbb{E}[M_t|\mathcal{F}_{\lambda,t}] = \mathbb{E}[g(x) - \mathbb{E}_{x \sim q_\lambda}[g(x)]|\mathcal{F}_{\lambda,t}] = 0$ where $\mathcal{F}_{\lambda,t} = \sigma(\lambda_i, M_i, i < t)$ is the filtration of $\lambda_t$ generated by the update trajectories. Then, we show that $M_t$ is square integrable. To illustrate this proof, notice that

$$||\nabla_\lambda L(\lambda) - g(x)||^2 = ||\nabla_\lambda L(\lambda)||^2 + ||g(x)||^2 - 2\langle \nabla_\lambda L(\lambda) \cdot g(x)\rangle.$$

We analyze the term separately. By the Cauchy-Schwarz inequality, we can obtain

$$-2\langle \nabla_\lambda L(\lambda) \cdot g(x)\rangle \leq 2||\nabla_\lambda L(\lambda)|| \cdot ||g(x)||$$
$$\leq ||\nabla_\lambda L(\lambda)||^2 + ||g(x)||^2.$$

Substitute the bound into the expanded squared norm:

$$||\nabla_\lambda L(\lambda) - g(x)||^2 \leq 2||\nabla_\lambda L(\lambda)||^2 + 2||g(x)||^2.$$

Then we apply Lipschitz and Boundedness Conditions, Using $||\nabla_\lambda L(\lambda)||^2 \leq K_1(1 + ||\lambda||^2)$ and $||g(x)||^2 \leq C$:

$$2||\nabla_\lambda L(\lambda)||^2 + 2||g(x)||^2 \leq 2K_1(1 + ||\lambda||^2) + 2C.$$

Since $1 \leq 1 + ||\lambda||^2$ for all $\lambda$, we have:

$$2K_1(1 + ||\lambda||^2) + 2C \leq (2K_1 + 2C)(1 + ||\lambda||^2).$$

Let $K = 2K_1 + 2C$. Therefore:

$$||\nabla_\lambda L(\lambda) - g(x)||^2 \leq K(1 + ||\lambda||^2).$$

Then, we can conclude that $M_t$ is square integrable. By combining these results, we could say that the update (40) is a stochastic approximation of the ODE (41) with a Martingale difference error term.

To show that the system converges to a locally stable point, we consider the following Lyapunov function,

$$\mathcal{L}(\lambda) = -L(q^*(\lambda), \lambda) + L(q^*(\lambda^*), \lambda^*), \tag{43}$$

where $\lambda^*$ is a local maximum point of the Langragian function. Consequently, there exists a neighborhood $\mathcal{B}_{\lambda^*}(r)$ centered at $\lambda^*$ with radius $r$, such that $\mathcal{L}(\lambda)$ is locally positive definite (i.e., $\mathcal{L}(\lambda) \geq 0$) for all $\lambda$ within this neighborhood. Since it is a local maximum point, we can obtain that

$$\frac{d\lambda_t}{dt} = [\nabla_\lambda L(q^*, \lambda]_+|_{\lambda=\lambda^*} = 0. \tag{44}$$

which means that $\lambda^*$ is a stationary point. Then we calculate the time-derivative of the Lyapunov function,

$$\frac{d\mathcal{L}(\lambda_t)}{dt} = -\nabla L_{\lambda_t}(\lambda_t)[\nabla L_{\lambda_t}(\lambda_t)]_+ \leq -||[\nabla L_{\lambda_t}(\lambda_t)]_+||^2 \leq 0 \tag{45}$$

which is nonzero when $[\nabla L_{\lambda_t}(\lambda_t)]_+ \neq 0$. Based on the Lyapunov theory, $\lambda_t$ will finally converge to $\lambda^*$.

Therefore, under conditions above, iterates $\lambda_t$ converge to the set of stable equilibrium points of the ODE (41), i.e., $\lambda^*$, which is the local maximum point of $L(q^*(\lambda), \lambda)$.

**Step 3 (Local Saddle Point)** After the steps above, we could conclude that $q^*$ is the minimum point of the $L(q, \lambda^*)$ over the feasible set. We now show that the equilibrium condition implies that the constraint satisfaction and the local saddle point for the original problem (9) are satisfied at $(q^*, \lambda^*)$. To complete the proof, we firstly show

$$\mathbb{E}_{x \sim q}[g(x)] \leq 0, \tag{46}$$

when $\lambda_t$ converges to $\lambda^*$. Suppose that $\mathbb{E}_{x \sim q}[g(x)] > 0$, this implies that $\frac{d\lambda_t}{dt} > 0$, which contradicts that $\frac{d\lambda_t}{dt}|_{\lambda_t = \lambda^*} = 0$. Therefore, the constraints are satisfied when dual update converges. Then we show that

$$\lambda \mathbb{E}_{x \sim q}[g(x)] = 0. \tag{47}$$

When the constraints are tight, we can easily show that the equation is satisfied. When $\mathbb{E}_{x \sim q}[g(x)] < 0$. Suppose that $\lambda^* \in (0, \lambda_{\max}]$, then there exists a small constant $\epsilon$ such that

$$\frac{d\lambda_t}{dt}|_{\lambda = \lambda^*} = \frac{1}{\epsilon}([\lambda^* - \epsilon \mathbb{E}_{x \sim q}[g(x)]]_+ - \lambda^*) = \mathbb{E}_{x \sim q}[g(x)] < 0 \tag{48}$$

which contradicts the stationary conditions. Then, we can obtain that

$$L(q^*, \lambda) \leq L(q^*, \lambda^*) \leq L(q, \lambda^*) \tag{49}$$

In conclusion, $(q^*, \lambda^*)$ is a local saddle point of $L(q, \lambda)$, i.e., $q^*$ is a local optimal solution to the original optimization problem (9)

## K   Analysis of Augmented Lagrangian Methods

In this section, we analyze the results of the Augmented Lagrangian methods. In the proposed methods, we introduce slack variables to handle inequality constraints, but the problem is still based on equality constraints. Therefore, in the analysis, we omit the slack variables and consider the case of equality constraints. We rewrite the equation (15),

$$L_A(q, \lambda, s) = \mathrm{KL}(q||p) + \lambda^{\mathsf{T}}[\mathbb{E}_{x_t \sim q}[g(x_t)]] + \frac{\rho}{2}\|\mathbb{E}_{x_t \sim q}[g(x_t)]\|^2 \tag{50}$$

Similar to the primal-dual methods, the optimal constrained distribution $q^*$ can be characterized as a tilted version of the original distribution $p(x) \propto e^{-f(x)}$, given by $q_\lambda^*(x)$. Sampling from $q_\lambda^*(x)$ can then be achieved by the update (16). Our analysis proceeds the assumption that the iterates are in a neighborhood of an optimal solution $(q^*, \lambda^*)$ that satisfies KKT conditions of original constrained optimization problem,

$$\frac{\delta L}{\delta q} = \log q(x) - \log p(x) + 1 + \lambda g(x) + \nu = 0 \tag{51}$$

$$\mathbb{E}_{x \sim q}[g(x)] = 0 \tag{52}$$

where $\nu$ is the term related to the normalization. The directions of time flows follow Remark 2.

At time step $t$ of the Augmented Lagrangian method, we have the following update rules,

$$x_{t+1} = x_t + \frac{\beta_t}{2}\left[\nabla_{x_t}\log p(x_t) - (\lambda_t + \rho_t(\mathbb{E}_{x_t \sim q}[g(x_t)]))^T\nabla_{x_t}g(x_t)\right] + \sqrt{\beta_t}z_t, \tag{53}$$

When we have fixed $\bar{\lambda}_t$ and $\bar{\rho}_t$, following the same assumptions in the proof of convergence of primal update for primal-dual methods, we could conclude that the distribution of $x_t$ generated by (53) converges to the unique stationary distribution $q_{\bar{\lambda}_t, \bar{\rho}_t}^*$, which is the optimal solution to the problem (9) with fixed $\bar{\lambda}_t$ and $\bar{\rho}_t$.

$$q_{t+1} = q_{\bar{\lambda}_t, \bar{\rho}_t}^* = \arg\min_q L_A(q; \bar{\lambda}_t, \bar{\rho}_t) \tag{54}$$

In the neighborhood of the optimal solution, the (53) approximates the related optimal solution, which is analogous to the first-order approximation in optimization methods. Since the solution is optimal, we have the following first-order optimality conditions $\delta L_A/\delta q = 0$, stating that the optimal distribution $q_{t+1}(x)$ for this subproblem satisfies:

$$\log q_{t+1}(x) - \log p(x) + 1 + \lambda_t g(x) + \rho_t \mathbb{E}_{x \sim q_{t+1}}[g(x)]g(x) + \nu = 0 \tag{55}$$

where $\nu$ is the term related to the normalization. This implies that $q_{t+1}(x)$ has the form:

$$q_{t+1}(x) \propto p(x)\exp\left(-\left(\lambda_t + \rho_t(\mathbb{E}_{x_t \sim q}[g(x_t)])\right)g(x)\right) \tag{56}$$

Following the solution of the Langevin update to obtain $q_{t+1}$, the Lagrange multiplier estimate is updated as follows:

$$\lambda_{t+1} = \lambda_t + \rho_t \mathbb{E}_{x \sim q_{t+1}}[g(x)] \tag{57}$$

Then we substitute (57) into (55) and obtain

$$\log q_{t+1}(x) - \log p(x) + 1 + \lambda_{t+1} g(x) + \nu = 0 \tag{58}$$

This form is identical to the KKT condition for the original constrained optimization problem. For convergence, we need $\lambda_{t+1} \to \lambda^*$ and, that the constraints are satisfied, i.e., $\mathbb{E}_{x \sim q_{t+1}}[g(x)] \to 0$. If the penalty parameter sequence $\rho^{(k)}$ is chosen such that $\rho^{(k)} \to \infty$, the solution to the optimization problem will make $\mathbb{E}_{x \sim q}[g(x)] \to 0$. Furthermore, based on the dual update rule, $\lambda_t$ will converge to $\lambda^*$ when bounded. Otherwise, the augmented Lagrangian will goes to infinity. According to Proposition 2.4 with Assumption S in [49], when we have conditions on twice differentiability and linear independence constraint qualification, we could have a rigorous convergence result in augmented Lagrangian methods. Given these conditions, as $t \to \infty$, $q_t(x) \to q^*(x)$, $\lambda_{t+1} \to \lambda^*$, $\mathbb{E}_{x \sim q^*}[g(x)] \to 0$,

Thus, the sequence of distributions $q_t$ generated by the iterative process converges to the optimal distribution $q^*$ that minimizes $\mathrm{KL}(q\|p)$ subject to $\mathbb{E}_{x \sim q}[g(x)] = 0$. The penalty term $\frac{\rho}{2}\|\mathbb{E}_{x \sim q}[g(x)]\|^2$ ensures that for large $\rho$, any significant constraint violation heavily penalizes $L_A$, driving iterates towards feasibility.

## L    Clarifications on DCBF constraints

Unlike a static state constraint, the DCBF constraint at each state $x^\tau$ depends on its neighboring state $x^{\tau+1}$. This coupling means that the gradient guidance at each trajectory point varies according to the states of adjacent points. In our three methods, the implementation differs as follows: for the Projected Method, the projection step solves a constrained optimization problem where the feasible set is defined by all coupled DCBF inequalities simultaneously enforced along the entire trajectory. For the Primal-Dual Methods, the update rule should be written as follows:

$$x_{t-1}^\tau = x_t^\tau + \frac{\beta_t}{2} \left( \nabla_{x_t^\tau} \log p(x_t^\tau) + (\lambda_t^{\tau-1})^T \nabla_{x_t} h(x_t) - (\lambda_t^\tau)^T (1-\alpha) \nabla_{x_t^\tau} h(x_t^\tau) \right) + \sqrt{\beta_t} \epsilon_t,$$

$$\lambda_{t-1}^\tau = \left[ \lambda_t^\tau - \eta_\lambda E_{x \sim q}[h(x_t^{\tau+1}) - (1-\alpha)h(x_t^\tau)] \right]_+,$$

For ALM, the update rule should be written as follows:

$$x_{t-1}^\tau = x_t^\tau + \frac{\beta_t}{2} \Bigg[ \nabla_{x_t^\tau} \log p(x_t^\tau) - (\lambda_t^{\tau-1})^T \nabla_{x_t^\tau} h(x_t^\tau) - \rho_t(E_{x \sim q}[h(x_t^\tau) - (1-\alpha)h(x_t^{\tau-1})]$$

$$- s_t^{\tau-1})^T \nabla_{x_t^\tau} h(x_t^\tau) + \lambda_t^\tau (1-\alpha)^T \nabla_{x_t^\tau} h(x_t^\tau) + \rho_t(1-\alpha)(E_{x \sim q}[h(x_t^{\tau+1})$$

$$- (1-\alpha)h(x_t^\tau)] - s_t^\tau)^T \nabla_{x_t^\tau} h(x_t^\tau) \Bigg] + \sqrt{\beta_t} z_t,$$

$$s_t^{\tau-1} = \left[ E_{x \sim q}[h(x_{t+1}^\tau) - (1-\alpha)h(x_t^\tau)] + \lambda_t^\tau/\rho_t \right]_+,$$

$$\lambda_{t-1}^\tau = \lambda_t^\tau + \rho_t(E_{x \sim q}[h(x_t^{\tau+1}) - (1-\alpha)h(x_t^\tau)] - s_{t-1}^\tau).$$

## M    The Inverse Dynamics Model (IDM)

We provide a detailed description and discussion of the Inverse Dynamics Model (IDM) used in our framework, including its rationale, architecture, and training methodology.

### M.1    Rationale and Training from Expert Data

In our framework, we assume that while an inverse dynamics model exists for the system, it is not accessible in a closed form derived from first-principle physics. The forward dynamics are treated as a black box, making classical model-based inversion infeasible.

To overcome this, we learn the IDM from data. Both the diffusion model and the IDM are trained offline using a dataset of expert trajectories. These trajectories are collected from interactions with the true physical system and are therefore composed entirely of dynamically feasible state-action

sequences. By training on this data, the IDM learns a valid mapping from state transitions $(x^\tau, x^{\tau+1})$ to the actions $(u^\tau)$ that produce them.

Once trained, this neural network-based model is "available" during deployment. It can efficiently convert the safe state trajectories generated by our constrained diffuser into the corresponding actions through a single forward pass, ensuring both state-action consistency and high computational efficiency.

## M.2    Model Architecture and Consistency

We implement the IDM as a deterministic model using multilayer perceptrons (MLPs). To enhance stability, we discretize the continuous action space into a finite number of bins, transforming the regression task of predicting an action into a more stable classification task.

The model's architecture is autoregressive, generating the action vector dimension by dimension. It first encodes the concatenated initial and next states $(x^\tau, x^{\tau+1})$ into a latent feature vector. Then, for each dimension of the action vector, a dedicated MLP predicts its value based on both the state feature vector and an embedding of all previously generated action dimensions. This autoregressive structure allows the model to capture dependencies between action dimensions, leading to more coordinated and realistic actions. The specific implementation follows the methodology presented in [4].

To ensure the model learns a smooth and consistent mapping within the expert data distribution, consistency can be further encouraged by adding a regularization term to the loss function during training. This term typically penalizes large action magnitudes, promoting minimum-effort solutions that generalize well and avoid erratic behavior.

## M.3    Dynamic Feasibility and Future Work

A key challenge, particularly for underactuated systems, is the risk that the planner could generalize outside the expert data distribution and propose a kinematically impossible transition. While our current experiments focus on state-based constraints to validate the core framework, our method is general and can be extended to handle constraints related to dynamic feasibility.

Specifically, we can enforce constraints directly on the output of the IDM during the reverse diffusion process, ensuring that the planner only generates trajectories that are executable within the system's actuation limits. This ensures all planned states are reachable, assuming the learned IDM is accurate.

While ensuring the existence of control actions for underactuated robots is a critical topic, a full exploration is beyond the scope of this paper. We plan to address this in future work, which will also explore more expressive and efficient IDM architectures, such as Neural ODE–based models, to better capture complex continuous-time dynamics.

# N    Visualization of Diffusion Process

The following visualizations show how trajectories evolve during denoising under different constraint-handling strategies.

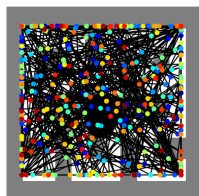 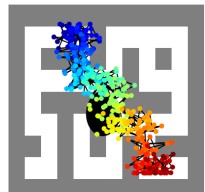 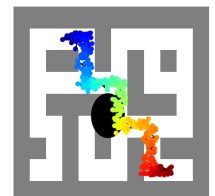 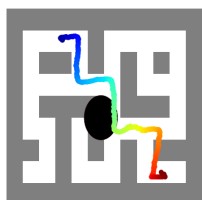

Figure 6: Diffusion process for vanilla diffusion

**Vanilla Diffusion:** The trajectory is recovered from Gaussian noise through stepwise denoising over the entire trajectory. No constraints are enforced and paths may intersect obstacles or violate safety conditions.

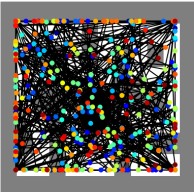 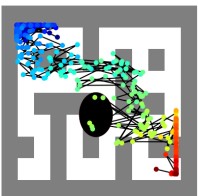 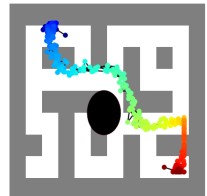 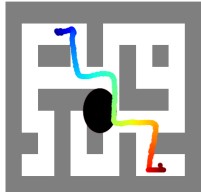

Figure 7: Conditional Diffusion

**Conditional Diffuser:** Each denoising step is guided by the gradient of a classifier. Early steps resemble vanilla diffusion, while later steps steer entire trajectories away from obstacles, producing safer paths at the cost of potential deviation from expert-like behavior.

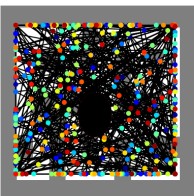 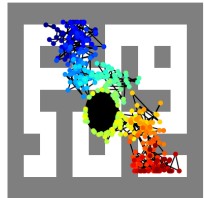 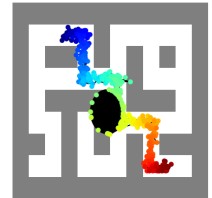 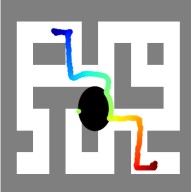

Figure 8: Diffusion process for SafeDiffuser

**SafeDiffuser:** If any point enters a constraint-violating region during denoising, that step is adjusted to move the point directly to the safe boundary. This ensures all points remain within or on the constraint boundary at every step while leading to some local traps.

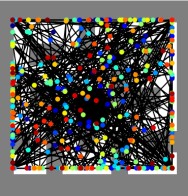 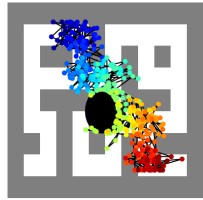 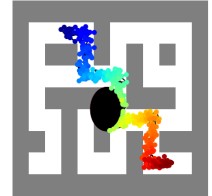 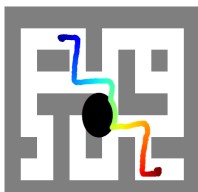

Figure 9: Diffusion process for projected methods

**Projected methods:** Similar to SafeDiffuser, but applies a geometric projection operator that any violating point is immediately projected onto the nearest constraint boundary.

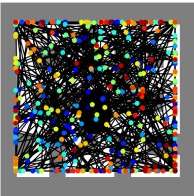 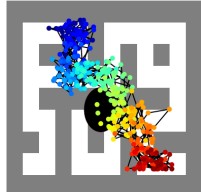 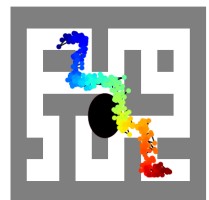 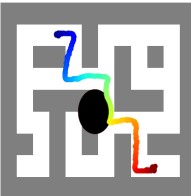

Figure 10: Diffusion process for primal-dual methods

**Primal-Dual methods:** Early denoising is close to vanilla diffusion. Near the end, only points violating constraints are gradually guided out, while safe points evolve like vanilla diffusion. This minimizes deviation from expert data while ensuring constraint satisfaction.

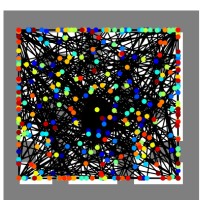 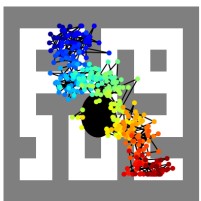 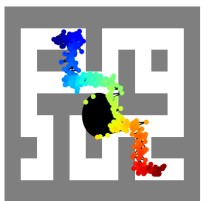 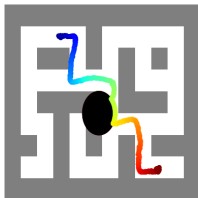

Figure 11: Diffusion process for ALM

**ALM:** Uses a quadratic penalty term to gradually guide violating points out during late denoising. This enables stronger constraint satisfaction.

