# OpenReview forum: "Constrained Diffusers for Safe Planning and Control"
_NeurIPS.cc/2025/Conference — NeurIPS 2025 poster_

### Official Review · Reviewer_SjPT · 2025-06-11

**Clarity:** 3
**Significance:** 4
**Originality:** 3
**Rating:** 5
**Confidence:** 3

**Summary:**

This paper introduces _Constrained Diffusers_, a framework that allows to enforce non-convex inequality constraints during the diffusion process. For enforcing such constraints, the authors propose three methods inspired by the constrained optimization literature: (1) projected diffusion sampling, (2) a primal-dual method, and (3) an Augmented Lagrangian Method (ALM). Practically, the constraints a formulated based on Discrete Control Barrier Functions which allows to enforce safety constraints for planning and control tasks. The method is evaluated on various tasks in simulation: Maze2D, locomotion (Hopper and Swimmer), and ball running. As baselines, and unconstrained (i.e., unsafe) diffusion model, SafeDiffuser, and Conditional Diffuser are considered. The results show that that some of the proposed techniques, such as the projected diffusion sampling, exhibit a good trade-off between task performance, limiting unsafe actions, and computational efficiency.

One important preface for this review: the manuscript that was available to the reviewer did not contain the appendix. Therefore, the reviewer was not able to fully assess all technical details and experimental results. The review is based on the main manuscript only.

**Questions:**

- What are the limitations of the proposed approach(es)?
- The authors do not discuss how the Diffuser models are trained. Are they trained from scratch or are you adopting pre-trained models? If the former, how are the demonstrations provided in the given environments (e.g., Maze2D, Hopper, Swimmer, PybulletBallRunning) If the latter, which pre-trained models are used?
- How can enforcing such constraints during the diffusion process be ("inuitively") understood? Can you show a visualization that shows diffusion of the trajectories as done by the vanilla diffusion model and how such trajectories are diffused differently when the constraints are enforced? Please aim to do so for all three constrained optimization methods that are proposed in this paper and also, preferably, for the baselines (SafeDiffuser and Conditional Diffuser).
- The SafeDiffuser baseline is only (fully) evaluated on the Maze2D task. Why is this the case? The reviewer does not understand the commeent in line 238 of page 8 that states "SafeDiffuser methods applies only to the planning stage so we do not compare their metrics in the implementation stage." Why is this the case? The reviewer would expect that the SafeDiffuser method can be applied to all tasks in a similar way as the proposed methods. Also, why is the SafeDiffuser not evaluated on the PybulletBallRunning task? The reviewer would expect that the SafeDiffuser can be applied to this task as well.
Most of these questions relate to the clarity and presentation of the paper and if the authors addresses these questions in the rebuttal and the revised version of the paper, the reviewer would be happy to increase the score.
- How are the Discrete Control Barrier Functions $h(x)$ transformed into the diffusion constraints $g(\cdot)$. Is this just a simple reshuffling of terms in the inequality equation? Or is there more to it? This seems like a crucial detail that is ommited in the paper.
- How is $x_t$/$x_0$ defined? Does it contain the entire trajectory? If yes, over which horizon? In Algorithm 1, the authors also use the variables $x_0^0$ and $x_0^1$ but they are not defined. How do they related to $x_0$? Also, practically speaking, how is the constrained diffusion policy used during inference? Does re-planning occur at every time step (i.e., the policy predicts an entire trajectory but only the first action is executed and then the policy is re-evaluated at the next time step)? Or does the policy predict a single action at every time step? The reviewer would like to see a more detailed description of how the constrained diffusion policy is used during inference.
- Contribution 2 (lines 53-56 of page 2): The reviewer questions why contribution 2 is a contribution. What exactly is the contribution here? Is it to use Control Barrier Functions in the context/as constraints of diffusion models? If so, this is not a new contribution as there has been prios work on this (e.g., SafeDiffuser). To the reviewer, it just seems that the authors use CBFs as a particular choice for generating the constraints $g(\cdot)$, which seems like a reasonable choice but not a contribution in itself. The reviewer suggests that the authors clarify this point in the paper.

**Ethical Concerns:**

["NO or VERY MINOR ethics concerns only"]

**Final Justification:**

The authors have been very responsive during this interactive discussion period, and their comprehensive replies helped answer the questions and address the concerns raised by the reviewers. Furthermore, the authors have added additional results and ablation studies to better support the claims made in the paper. The reviewer greatly appreciates the authors for their work and communication during this discussion period. Finally, the reviewer also greatly values the timely topic of this paper and the contributions made that allow for a new approach for integrating safety constraints into (robot) diffusion policies.

Therefore, the reviewer recommends this paper for acceptance and adjusts their score to a "5: Accept".

However, as the discussion period with all reviewers has shown, the "devil lies in the details" and many of these details, contributions, etc., were not clear in the initially submitted manuscript. Therefore, the reviewer strongly recommends that the authors carefully revise their manuscript and ensure that these elements, questions, and contributions become **much** clearer in the final paper.

**Limitations:**

The authors do not discuss the limitations of their approach in the main text, although they state that the limitations are discussed in the appendix to which the reviewer did not have access. In any case, the reviewer suggest that the authors should include a concise discussion of the limitations in the main text.

**Paper Formatting Concerns:**

The paper manuscript does not contain the appendix which is referenced various times in the main text. The reviewer questions if this submission can be properly reviewed without the appendix.

**Quality:**

3

**Strengths And Weaknesses:**

## Strengths

- The paper's contribution seems to be novel and valuable for the community and takes a step towards integrating safety constraints into state-of-the-art diffusion models.
- The paper includes a comprehensive benchmarking between various methods of enforcing constraints and a comparison to existing baselines, such as SafeDiffuser and Conditional Diffuser.

## Weaknesses

- The paper's presentation and clarity could be improved. The reviewer found it to be somewhat difficult to follow how all the pieces and components touched in this paper fit together. Furthermore, illustrations of how the different constrained optimization methods modulate the vanilla difffusion process are missing and would be helpful. The reviewer suggest that the authors rewrite the paper to improve clarity and presentation. At least the section that describes the overall idea and approach of the paper should be relatively easy to follow (also in terms of notation) and should not require the reader to have a deep understanding of all the concepts that are touched in this paper.
- The paper's discussion of the related work is not sufficient and appears vague and hand-wavy. Specifically, the reviewer is referring here to lines 26-33 of the introduction. The reviewer excepts a clear discussion about the limitations of the existing methods in this area (e.g., SafeDiffuser, Conditional Diffuser) and how the proposed approach overcomes these limitations.
- The paper does not include real-world experiments, which seems acceptable though given the focus and contributions of the paper.

### Detailed Comments on Minor Issues

- Page 3, line 81: "... by learning the noise to denoise..." Is this a typo?
- Equation (4): From this point, you use $z$ all of the sudden without the subscript $t$ (i.e., instead of $z_t$) without redefining it. Please clarify.
- Maybe Figure 2 should be Figure 1? The current Figure 2 seems suitable to communicate the overall idea of the method and should be the first figure in the paper.
- Tables 1-3: Please cite the baselines in the tables.
- Table 1: why is the "Time/s" result for "Projected" in "maze2d-umaze" marked in bold? There are other methods that are equally fast or even slightly faster that are not marked in bold.
- Figure 4: Please use the same order for the methods in the legend as in the tables. Also, I am not able to visually distinguish the "Projected" method (in purple) from the other methods. Please improve the presentation of the figure.

---

> ### Author Rebuttal · Authors · 2025-07-31
>
> We thank the reviewer for the detailed comments and valuable feedback. We appreciate your recognition of the novelty and significance of our work. We would like to clarify that the appendix is included in the supplementary materials, packaged with our code as a single ZIP file. If there are any remaining access issues, we are happy to provide the appendix text during the discussion phase.
>
> **W1: Improve the presentation and clarification**
>
> **Response:** We thank the reviewer for their constructive feedback. In the revision, we will reorganize the introduction and methods to present a clear overview and refine the notation. We plan to put the vanilla diffusion process before the constrained optimization methods to clearly show the connections.
>
> **W2: Discussion on the limitations of the existing method**
>
> **Response:** We will explain the limits of existing methods to motivate our approach in the revised paper. Conditional Diffusers work well for qualitative constraints like “grab the purple box,” but have trouble with strict safety rules like avoiding collisions. Their soft conditioning can’t always enforce these rules exactly, causing violations. SafeDiffuser uses CBF to enforce hard safety constraints but needs to solve a slow optimization problem at every step, which hurts real-time performance. Also, SafeDiffuser changes states during diffusion but not actions, causing a mismatch that may break safety during online execution.
>
> **W3: Does not include real-world experiments**
>
> **Response:** We thank the reviewer for their comment and understanding of our paper’s scope. This work focuses on establishing the Constrained Diffusers framework and validating its efficiency in some standard environments. We will extend to this in the future.
>
> **W4: Detailed Comments on Minor Issues**
>
> **Response:** 1. We revised it to "...by learning the noise via neural networks." 2. $z_t \sim \mathcal{N}(0, I)$ is standard Gaussian noise. 3. The reviewer is correct that Figure 2 shows the overall framework, and we will rearrange the order. 4. We will cite the baselines in the tables. 5. We marked the "Time/s" result for "Primal Dual" in "maze2d-umaze" rather than "Projected". 6. We will mark it in order, and make each line clearer.
>
> **Q1: Limitations of the proposed approaches**
>
> **Response:** We summarize the limitations into theoretical and practical ones. On the theoretical end, our methods rely on assumptions including inaccurate score function estimation. On the practical end, we made an assumption that we have access to constraints, and lack robustness analysis. For detailed analysis, please refer to Section A in the Appendix from Supplementary materials.
>
> **Q2: How the Diffuser models are trained**
>
> **Response:** All diffuser models in our experiments were trained from scratch using expert demonstration data. For Maze2D, we used Maze2D-umaze-v1 and Maze2D-large-v1 from D4RL; for MuJoCo locomotion tasks, we used hopper-medium-expert-v2 from D4RL and swimmer-medium-expert-v2 from Minari; for pybullet ball running tasks, we used SafetyBallRun-v0 from DSRL.
>
> **Q3: How to understand diffusion process when enforcing such constraints?**
>
> **Response:** Since figures cannot be included in this rebuttal, we provide a brief explanation using the umaze environment as an example. Detailed figures will be added in the revised paper.
> - Vanilla Diffusion: The trajectory is recovered from Gaussian noise through stepwise denoising over the entire trajectory.
> - Conditional Diffusion: Each denoising step is guided by the gradient of a classifier. Early steps resemble vanilla diffusion, while later steps steer entire trajectories away from obstacles, producing safer paths.
> - Safe Diffuser and Projected: If a point enters a constraint-violating region during denoising, that step is adjusted to move or project to the point directly to the safe boundary, ensuring all points remain within or on the constraint boundary at every step.
> - PD: Similar to conditional diffusion, early denoising is close to vanilla diffusion. Near the end, only points violating constraints are gradually guided out, while safe points evolve like vanilla diffusion, minimizing deviation from expert data.
> - ALM: Uses a quadratic penalty term to gradually guide violating points out during late denoising, enabling smoother corrections and stronger constraint satisfaction.
>
> **Q4: Fully evaluate SafeDiffuser on all tasks**
>
> **Response:** Thank you for this observation. he problem comes from a mismatch between states and actions in the original SafeDiffuser when moving from planning to online execution. SafeDiffuser uses CBF during reverse diffusion to create a safe state trajectory in planning. However, the actions used in execution are not based on this planned safe state. The model is trained on both states and actions, but during diffusion, only the states are guided, and actions are sampled directly without adjustment. This leads to a mismatch where the executed actions may not follow the planned safe states. Therefore, the original method is mainly suitable for planning, and online execution results were not reported. To compare fairly, we ran additional experiments combining SafeDiffuser with our IDM on all tasks. The results show SafeDiffuser works well in planning but performs less well online. We will include them in the revised paper.
>
> |Algorithm|Env|Reward|PlanningViolations|Impl.Violations|ViolationRates(%)|Time/s|
> |-|-|-|-|-|-|-|
> |SafeDiffuser(IDM)|Hopper|3514±46|0±0|0.28±0.06|0.27±0.05|0.9307|
> ||Swimmer|60.4±6.1|0±0|0.65±0.21|0.97±0.25|0.9301|
> ||SafetyBallRunning|820.8±29.8|--|1.13±1.08|21.9±12.8|2.6e-2|
>
> **Q5: How are the DCBF transformed into the diffusion constraints**
>
> **Response:** Thank you for the feedback. You are correct that unlike a static state constraint, the DCBF constraint at each state$x^\\tau$ depends on its neighboring state $x^{\\tau+1}$. This coupling means that the gradient guidance at each trajectory point varies according to the states of adjacent points. In our three methods, the implementation differs as follows: for the Projected Method, the projection step solves a constrained optimization problem where the feasible set is defined by all coupled DCBF inequalities simultaneously enforced along the entire trajectory. For the Primal-Dual Methods, the update rule should be written as follows:
>
> $$
> {x_{t-1}^\\tau} = {x_{t}^\\tau} + \frac{\beta_t}{2}\\left({\\nabla _{x_t^\\tau}}\\log p(x_t^\\tau)- (\\lambda_t^{\\tau-1})^T{\\nabla _{x_t^\\tau}} h(x_t^\\tau)+(\\lambda_t^{\\tau})^T(1-\\alpha){\nabla _{x_t^\\tau}} h(x_t^\\tau)\right) + \\sqrt \\beta_t  {{z}_t},
> $$
>
> $$
> \\bf{\\lambda}_{t-1}^\\tau = \\left[\\bf{\\lambda}_t^\\tau + {\\eta_{\\bf{\\lambda}}}\\mathbb{E}_{x \\sim q}[h(x_t^{\\tau+1})-(1-\\alpha)h(x_t^\\tau)]\\right]_+.
> $$
>
> For ALM, the update rule should be written as follows:
>
> $$
> {x_{t-1}^\\tau} = {x_{t}^\\tau} + \\frac{\\beta_t}{2}\\left[\\nabla_{x_t^\\tau}\log p(x_t^\\tau) - (\\lambda_{t}^{\\tau-1} + \\rho_t\\left(\\mathbb{E}_{x_t \\sim q}[h(x_t^{\\tau})-(1-\\alpha)h(x_t^{\\tau-1})] + s_{t}^{\\tau-1}\\right) \\
> -\\lambda_{t}^{\\tau}(1-\\alpha) - \\rho_t(1-\\alpha)\\left(\\mathbb{E}_{x_t \\sim q}[h(x_t^{\\tau+1})-(1-\\alpha)h(x_t^\\tau)] + s_t^\\tau\\right))^T\\nabla_{x_t} g(x_t)\\right] + \\sqrt{\\beta_t}{z_t},
> $$
>
> $$
> s_t^\\tau = \\left[-\\mathbb{E}_{x_t \\sim q}[h(x_t^{\\tau+1})-(1-\\alpha)h(x_t^\\tau)] - \\lambda_t^\\tau / \\rho_t\\right]_+,
> $$
>
> $$
> \\lambda_{t-1}^\\tau = \\lambda_{t}^{\tau + \\rho_t\\left(\\mathbb{E}_{x_t \\sim q}[h(x_t^{\\tau+1})-(1-\\alpha)h(x_t^\\tau)] + s_t^\\tau \\right).
> $$
> We will add this in the revised paper.
>
> **Q6: Clarification on notations and description on how the constrained diffusion is used during inference**
>
> **Response:** We first clarify that our paper involves two time horizons: the diffusion process horizon denoted by $t$ and $T$, and the physical environment horizon denoted by $\\tau$ and $\\mathcal{T}$.
> - $x_0$ represents the entire clean trajectory over the planning horizon $\\mathcal{T}$ at diffusion timestep 0.
> - $x_t$ denotes the noisy version of this trajectory at diffusion step $t$.
> - The notation $x_0^\\tau$ refers to the $\\tau$-th state in $x_0$, where $x_0^0$ is the current agent state, and $x_0^1$ is the next planned state.
> During inference, our method proceeds as follows:
> 1. The agent observes its current state, which becomes $x_0^0$.
> 2. A full reverse diffusion process is run from noise to generate $x_0$. The constrained sampling method ensures safety state constraints are satisfied.
> 3. The IDM computes the control $u^\tau$ for the safe state transition $x_0^0 \\to x_0^1$, which is executed in the environment.
> 4. The remaining trajectory is discarded. After the environment updates to the next state, the process repeats from Step 1.
> Thus, the policy predicts a full safe trajectory at each time step but executes only the first action.
>
> **Q7: Clarification on Contribution 2**
>
> **Response:** We will rephrase it to "We incorporate the DCBF to ensure that pairs of consecutive states are safe and dynamically feasible, allowing the IDM to find valid actions during online execution." To clarify, our method leverages the iterative diffusion process to enforce DCBF constraints efficiently, avoiding the heavy computation of traditional QP-based CBF methods. Compared to static constraints, DCBF ensures forward invariance, making the trajectory dynamically feasible and more reliable during online execution. Unlike SafeDiffuser, which solves a costly quadratic program at each diffusion step, we focus on safety between consecutive states in the planned trajectory, guaranteeing dynamic feasibility. Instead of solving QPs, we guide the reverse sampling using gradients of the DCBF, gradually pushing noisy trajectories toward clean, feasible ones. Therefore, we believe that our approach provides an efficient and practical way to generate safe, dynamically feasible trajectories for online execution.

---

> > ### Author Response · Authors · 2025-08-05
> > **Clarification on our rebuttal**
> >
> > Dear Reviewer,
> >
> > Thank you again for your time and for the insightful suggestions that helped us improve our work.
> >
> > We recently noticed that some of the LaTeX formulas in our rebuttal were not properly presented in the system, which may have affected readability. To clarify our response, we are providing a corrected version with properly displayed equations for your reference.
> >
> > For the Primal-Dual Methods, the update rule should be written as follows:
> >
> > $$
> > {x_{t-1}^\\tau} = {x_{t}^\\tau} + \\frac{\\beta_t}{2}\\left({\\nabla _{x_t^\\tau}}\\log p(x_t^\\tau)+ (\\lambda_t^{\\tau-1})^T{\\nabla _{x_t^\\tau}} h(x_t^\\tau)- (\\lambda_t^{\\tau})^T(1-\\alpha){\\nabla _{x_t^\\tau}} h(x_t^\\tau)\\right) + \\sqrt \\beta_t  {{z}_t},
> > $$
> >
> > $$
> > {{\\bf{\\lambda }}_ {t-1}^\\tau} = \\left[{{\\bf{\\lambda}}_ {t}^\\tau} - {\\eta _ {\\bf{\\lambda }}}\\mathbb{E}_ {x \\sim q}[h(x_t^{\\tau+1})-(1-\\alpha)h(x_t^\\tau)]\\right]_+.
> > $$
> > For ALM, the update rule should be written as follows:
> >
> > $$
> > {x_{t-1}^\\tau} = {x_{t}^\\tau} + \\frac{\\beta_t}{2}\\left[\\nabla_{x_t^\\tau}\\log p(x_t^\\tau) - (\\lambda_{t}^{\\tau-1})^T\\nabla_{x_t} h(x_t) - \\rho_t\\left(\\mathbb{E}_ {x_t \\sim q}[h(x_t^{\\tau})-(1-\\alpha)h(x_t^{\\tau-1})]-s_{t}^{\\tau-1}\\right)^T\\nabla_{x_t} h(x_t)
> > +\\lambda_{t}^{\\tau}(1-\\alpha)^T\\nabla_{x_t} h(x_t) + \\rho_t(1-\\alpha)\\left(\\mathbb{E}_ {x_t \\sim q}[h(x_t^{\\tau+1})-(1-\\alpha)h(x_t^\\tau)]-s_t^\\tau\\right)^T\\nabla_{x_t} h(x_t)\\right] + \\sqrt{\\beta_t}{z_t},
> > $$
> >
> > $$
> > s_{t-1}^\\tau = \\left[\\mathbb{E}_ {x_t \\sim q}[h(x_t^{\\tau+1})-(1-\\alpha)h(x_t^\\tau)] + \\lambda_t^\\tau/\\rho_t\\right]_+,
> > $$
> >
> > $$
> > \\lambda_{t-1}^\\tau = \\lambda_t^\\tau + \\rho_t\\left(\\mathbb{E}_ {x_t \\sim q}[h(x_t^{\\tau+1})-(1-\\alpha)h(x_t^\\tau)] - s_{t-1}^\\tau \\right).
> > $$
> >
> > We will add this in the revised paper.
> >
> > If you have any further questions or require additional clarification on our work, we would be happy to provide further explanations.
> >
> > Best regards,
> > Authors

---

> > > ### Author Response · Authors · 2025-08-06
> > > **Follow up on our rebuttal**
> > >
> > > Dear Reviewer,
> > >
> > > Thank you again for your time and for the insightful suggestions that helped us improve our work.
> > >
> > > As the discussion period is coming to a close, we would like to kindly check whether our previous response addressed your concern, or if you have any further points you would like to discuss. We are happy to provide any additional clarifications if needed.
> > >
> > > Best regards,
> > >
> > > Authors

---

> > ### Comment · Reviewer_SjPT · 2025-08-07
> > **Response to Rebuttal**
> >
> > First of all, sorry for the late response to the rebuttal. I find the reviewer workload to be quite high and it is challenging to keep up with all the reviews and rebuttals.
> >
> > I would like to thank the authors for their detailed and constructive response to the initial review. As already mentioned in the initial review, the reviewer appreciates the timely motivation and the contributions made in this paper. The clarifications answered many of my and other reviewers' questions. In particular, I appreciate the additional experiments and ablation studies that were presented in the rebuttal. In general, the reviewer is happy with the rebuttal by the authors. A few open questions remain - some of which are addressed below and some of which are posted in responses to other reviewers.
> >
> > In particular, while the differences with respect to SafeDiffuser are now (a bit) more clear, I am still not having a clear overview of all the differences and some of the authors responses seemed a bit confusing and even contradicting. I would appreciate if the authors could add a comprehensive and detailed discussion of the differences between SafeDiffuser and all the methods proposed in this paper.
> >
> > ## Responses
> >
> > >  We appreciate your recognition of the novelty and significance of our work. We would like to clarify that the appendix is included in the supplementary materials, packaged with our code as a single ZIP file. If there are any remaining access issues, we are happy to provide the appendix text during the discussion phase.
> >
> > Sorry, this was my bad for not checking the ZIP file. There appears to be some inconsistency this years as some papers have the appendix in the main PDF and others have it in a separate ZIP file.
> >
> > > Response: We summarize the limitations into theoretical and practical ones. On the theoretical end, our methods rely on assumptions including inaccurate score function estimation. On the practical end, we made an assumption that we have access to constraints, and lack robustness analysis. For detailed analysis, please refer to Section A in the Appendix from Supplementary materials.
> >
> > Sorry, for initially overseeing the statement of limitations in the appendix. I strongly suggest to add a short summary of the limitations in the main text, as it is important for the reader to be aware of them.
> >
> > > Response: Since figures cannot be included in this rebuttal, we provide a brief explanation using the umaze environment as an example. Detailed figures will be added in the revised paper.
> >
> > Thank you very much for providing the explanation. I very much look forward to the figures in the revised paper. I think this will significantly improve the clarity of the paper.
> >
> > > Response: Thank you for this observation. he problem comes from a mismatch between states and actions in the original SafeDiffuser when moving from planning to online execution. SafeDiffuser uses CBF during reverse diffusion to create a safe state trajectory in planning. However, the actions used in execution are not based on this planned safe state.
> >
> > Thank you for running these additional experiments ensuring a fair comparison. I look forward to seeing the results in the revised paper.
> >
> > > Q6: Clarification on notations and description on how the constrained diffusion is used during inference
> > >
> > > Response: We first clarify that our paper involves two time horizons: the diffusion process horizon denoted by  and , and the physical environment horizon denoted by  and .
> >
> > Thank you for this explanation. Please include such clarifications and details also in the revised paper.
> >
> > > The remaining trajectory is discarded. After the environment updates to the next state, the process repeats from Step 1. Thus, the policy predicts a full safe trajectory at each time step but executes only the first action.
> >
> > Have you tried out variations of this? Maybe executing five time steps at once before replanning and repeating the process?
> >
> > > Response: We will rephrase it to "We incorporate the DCBF to ensure that pairs of consecutive states are safe and dynamically feasible, allowing the IDM to find valid actions during online execution."
> >
> > Thank you for the explanation. However, the contribution of this work with respect to related work such as SafeDiffuser is still not clear to me when **only** reading your proposed rephrased 2nd contribution. I suggest to further rephrase the 2nd contribution to improve clarity.

---

> > > ### Author Response · Authors · 2025-08-08
> > > **Response to the reviewer's comment I**
> > >
> > > We sincerely appreciate the reviewer’s thorough evaluation and constructive feedback, and we fully understand the high workload involved in reviewing and rebuttals. We are glad that our previous responses have helped clarify many aspects of our work. As suggested, we will add a short summary of the limitations directly in the main text, in addition to the detailed discussion in the appendix. We will also include illustrative figures that visualize the diffusion process for each algorithm, clarifications on the comparison between our work and previous works, and additional results given in the rebuttal to improve the understanding.
> > >
> > > >Have you tried out variations of this? Maybe executing five time steps at once before replanning and repeating the process?
> > >
> > > R: We appreciate this excellent suggestion. In fact, we have explored such variations inspired by Model Predictive Control. For instance, in the Hopper environment, executing even three time steps at once often leads to degraded performance, and in some cases the hopper falls, causing task failure. This is partly due to the fact that the trajectories generated by diffusion models is a generalization of the expert trajectories, and the IDM may introduce additional errors. These factors can accumulate errors over multiple steps, reducing robustness. Therefore, in our current implementation, we adopt a one-step “closed-loop” execution strategy. Due to time constraints, it is difficult to conduct a complete evaluation experiment before the end of the discussion. But we are happy to include the corresponding experimental results in our revised paper to illustrate this effect. In future work, we also plan to further investigate, from a theoretical perspective, the trade-offs involved in executing multiple steps before replanning, in order to determine under what conditions such an approach may be optimal.
> > >
> > >
> > > > Rephrase the 2nd contribution to improve clarity.
> > >
> > > We agree that our second contribution, when stated in isolation, may not fully convey its significance and connection to our overall framework.
> > >
> > > The original thought for separating the contributions was to highlight two distinct novel aspects of our work when compared against different sets of prior literatures, which may regard as our second contribution:
> > >
> > > Compared to SafeDiffuser which has computational burden and is trying to make the diffusion process forward invariant, ours is more efficient and to make the physical trajectory forward invariant, ensuring the safety of the planned trajectory and enhancing the safety of the online implementation.
> > >
> > > Compared to traditional or imitation learning planners with CBF, our novelty lies in utilizing the iterative nature of the diffusion process. Our efficient PD and ALM algorithms enforce these dynamic CBF constraints using only simple gradient information, avoiding the significant computational bottleneck of traditional QP solvers.
> > >
> > > We also see that these two points are deeply interconnected and mutually enhancing. Therefore, to better represent our contribution, we propose merging them into a single, unified statement:
> > >
> > > "We propose Constrained Diffusers, a novel framework that enforces constraints on trajectories generated by pre-trained diffusion models, modifying reverse diffusion process through integrating constrained sampling methods: Projected, Primal-Dual and Augmented Lagrangian methods, avoiding costly retraining or model modifications during deployment. In addition, We introduce Discrete Control Barrier Functions defined on the physical trajectory, ensuring the generation of safe plans for online implementation and leverage the properties of iterative diffusion process to satisfy the constraints, avoiding the high computational cost of traditional methods."
> > >
> > > If you find this revised, unified phrasing to be clearer and more appropriate, we would be happy to adopt it in the revised manuscript.

---

> > > > ### Comment · Reviewer_SjPT · 2025-08-08
> > > >
> > > > > R: We appreciate this excellent suggestion. In fact, we have explored such variations inspired by Model Predictive Control. For instance, in the Hopper environment, executing even three time steps at once often leads to degraded performance, and in some cases the hopper falls, causing task failure. This is partly due to the fact that the trajectories generated by diffusion models is a generalization of the expert trajectories, and the IDM may introduce additional errors. These factors can accumulate errors over multiple steps, reducing robustness. Therefore, in our current implementation, we adopt a one-step “closed-loop” execution strategy. Due to time constraints, it is difficult to conduct a complete evaluation experiment before the end of the discussion. But we are happy to include the corresponding experimental results in our revised paper to illustrate this effect. In future work, we also plan to further investigate, from a theoretical perspective, the trade-offs involved in executing multiple steps before replanning, in order to determine under what conditions such an approach may be optimal.
> > > >
> > > > Thank you for this discussion and the reviewer would be happy if you could add a small ablation study on this topic to the final paper.
> > > >
> > > > > We also see that these two points are deeply interconnected and mutually enhancing. Therefore, to better represent our contribution, we propose merging them into a single, unified statement:
> > > > >
> > > > > "We propose Constrained Diffusers, a novel framework that enforces constraints on trajectories generated by pre-trained diffusion models, modifying reverse diffusion process through integrating constrained sampling methods: Projected, Primal-Dual and Augmented Lagrangian methods, avoiding costly retraining or model modifications during deployment. In addition, We introduce Discrete Control Barrier Functions defined on the physical trajectory, ensuring the generation of safe plans for online implementation and leverage the properties of iterative diffusion process to satisfy the constraints, avoiding the high computational cost of traditional methods."
> > > > >
> > > > > If you find this revised, unified phrasing to be clearer and more appropriate, we would be happy to adopt it in the revised manuscript.
> > > >
> > > > The reviewer understands that communicating the contributions of this work that (heavily) are based on technical in a single sentence/bullet point is quite challenging.
> > > >
> > > > The reviewer believes that the proposed changes are going in the right direction and is satisfied with the author's rebuttal.
> > > >
> > > > However, the reviewer suggest the authors to further iterate on the introduction and the statement of contributions. For example, when reading the proposed statement of contributions, I see the following deficiencies:
> > > >
> > > > - It would not be (immediately) clear which "traditional methods" you are referring to.
> > > > - I would argue that "Discrete Control Barrier Functions" are (probably) not something new by itself, but their application in this framework for inference is a new use case. The authors should rephrase to make this distinction more clear.

---

> > > ### Author Response · Authors · 2025-08-08
> > > **Response to the reviewer's comment II**
> > >
> > > > I would appreciate if the authors could add a comprehensive and detailed discussion of the differences between SafeDiffuser and all the methods proposed in this paper.
> > >
> > > Thank you for your suggestions.
> > >
> > > ### Diffusion with Constraints
> > >
> > > Both our work and SafeDiffuser address the same core challenge: a standard diffusion model used for imitation learning does not inherently handle strict mathematical safety constraints. In both cases, the goal is to ensure that the generated trajectory remains within a safe set defined by
> > > $$
> > > h(x^\tau) \ge 0, \quad \forall\tau\in\\{0,\dots,\mathcal{T}\\}.
> > > $$
> > > SafeDiffuser interprets the diffusion process from a control-theoretic perspective, where each denoising step $t$ is treated as a control action $u_t$ over the diffusion time horizon. To ensure safety, it solves a Control Barrier Function Quadratic Program (CBF-QP) at every intermediate diffusion step:
> > > $$
> > > \min_{u_t} \\|u_t - u_t^*\\|^2
> > > \quad \text{s.t.} \quad
> > > \dot{h}(x_t^\tau)u_t + \alpha h(x_t^\tau) \ge 0,
> > > \quad \forall\tau\in\\{0,\dots,\mathcal{T}\\},
> > > $$
> > > where the gradient $\dot{h}(\cdot)$ is computed based on the diffusion dynamics with respect to $t$ rather than the physical environment dynamics with respect to $\tau$. This ensures that each diffusion step remains in the safe set, but comes with a high computational cost because a full QP must be solved at every step (implemented in their code via the `qpth` Python package using a primal-dual interior point method).
> > >
> > > Our approach views the diffusion process as a constrained optimization problem and proposes three methods — Projected, Primal-Dual (PD), and Augmented Lagrangian Method (ALM). The Projected method is inspired by projected gradient descent and projected Langevin dynamics; instead of adjusting the control action $u_t$ as in SafeDiffuser, it directly projects the sampling points $x_t$ onto the feasible set:
> > > $$
> > > \min_{x} \\| x_t - x_t^* \\|^2
> > > \quad \text{s.t.} \quad
> > > h(x_t^\tau) \ge 0, \quad \forall\tau\in\\{0,\dots,\mathcal{T}\\}.
> > > $$
> > > This is an algorithmic distinction: we solve a QP to find the closest feasible sampling point, rather than a safe control update. The PD and ALM methods treat the constraint satisfaction problem over the entire diffusion trajectory as an optimization task, using efficient first-order gradient-based updates to enforce safety and thus avoiding the high cost of per-step QP solving.
> > >
> > > The above summarizes the difference between us and safediffuser at the algorithm level
> > >
> > > ### Online Implementation
> > >
> > > Furthermore, within our constrained diffuser framework, we adopt a Discrete-time Control Barrier Function (DCBF) defined over the entire **physical** trajectory:
> > > $$
> > > h(x^{\tau+1}) \ge (1 - \alpha) h(x^\tau),
> > > \quad \forall\tau\in\\{0,\dots,\mathcal{T}\\},
> > > $$
> > > which, to the best of our understanding, SafeDiffuser does not apply in this form. They directly apply hard constraints on the physical trajectories. Therefore, this is an additional part that we would like to realize online implementation safety, which decouples from both SafeDiffuser and our constrained diffusers on how to handle constraints in diffusion. We admit that SafeDiffuser can also use this as a constraint, but it will lead to more complicated calculations.
> > >
> > > We hope our response resolves your concerns. If you have any other questions, we would be happy to explain further for you.

---

> > > > ### Comment · Reviewer_SjPT · 2025-08-08
> > > > **Final Conclusion**
> > > >
> > > > The authors have been very responsive during this interactive discussion period, and their comprehensive replies helped answer the questions and address the concerns raised by the reviewers. Furthermore, the authors have added additional results and ablation studies to better support the claims made in the paper. The reviewer greatly appreciates the authors for their work and communication during this discussion period. Finally, the reviewer also greatly values the timely topic of this paper and the contributions made that allow for a new approach for integrating safety constraints into (robot) diffusion policies.
> > > >
> > > > Therefore, the reviewer recommends this paper for acceptance and will adjust their score accordingly.
> > > >
> > > > However, as the discussion period with all reviewers has shown, the "devil lies in the details" and many of these details, contributions, etc., were not clear in the initially submitted manuscript. Therefore, the reviewer strongly recommends that the authors carefully revise their manuscript and ensure that these elements, questions, and contributions become **much** clearer in the final paper.

---

> > > > > ### Author Response · Authors · 2025-08-09
> > > > > **Response to the reviewer's comment**
> > > > >
> > > > > We thank the reviewer for their constructive and encouraging feedback, as well as for updating the final rating and recommending our paper for acceptance.
> > > > >
> > > > > We agree with the reviewer’s points regarding the clarity of the Introduction and Statement of Contributions. In the revised version, we will carefully revise these sections.  We will clarify "traditional methods" in our revised paper (e.g., here traditional methods refer to traditional diffusion policies with safety and traditional QP based methods with CBF). We will also clarify that our contribution lies in introducing DCBF into this inference framework as a novel application for integrating safety constraints into robot diffusion policies, not DCBF itself.
> > > > >
> > > > > We will ensure that the final manuscript clearly conveys these details, so that the overall contributions are immediately understandable to readers.
> > > > >
> > > > > Thank you again for your valuable comment.

---

### Official Review · Reviewer_3UpD · 2025-06-12

**Clarity:** 3
**Significance:** 2
**Originality:** 2
**Rating:** 4
**Confidence:** 3

**Summary:**

This paper presented a constrained diffuser framework that does not rely on pre-training and is able to ensure safety when using the diffusion model for control and planning. The authors designed three different methods (ALM, projection, and Primal-Dual) to complete constrained sampling and satisfy the constraints represented by synthesized discrete control barrier functions. In the end, an inverse env model, which is supposed to be known a priori, is used to compute the action. By using this framework, the diffuser-based controller can have a good constraint satisfaction performance compared with existing baseline methods in various environments.

**Questions:**

See above

**Ethical Concerns:**

["NO or VERY MINOR ethics concerns only"]

**Final Justification:**

My concerns have been resolved.

**Limitations:**

No societal impact relevant to this work

**Paper Formatting Concerns:**

No paper formatting concerns

**Quality:**

2

**Strengths And Weaknesses:**

Strengths:
1. The paper is written clearly will good illstration. Fig 1 is pretty intuitive and can demonstrate the performance of different methods. Fig 2 is  easy to understand and is able to help the audience understand the gist of the framework quickly.
2. The topic is important and interesting. The method proposed in this paper is easy to use since no additional training is required.
3. A time-varying situation is considered important but relatively less studied.

Weakness:
1. The benchmark method SafeDiffuser is discussed and compared in the experiments where the environments are not changed. However, the performances in time-varying scenarios are not shown.
2. The author used an inverse dynamical model to generate the control from the state transition. However, in many dynamical models, there could be different action solutions from the inverse dynamical model. It would be nice if the author could elaborate on how to guarantee the consistency of the controls due to the significance in practice.
3. The constrained sampling framework for safe control/planning is not a new idea and has been extensively studied and applied in areas like safe RL for robotics. It would be better if the authors could summarize those works and draw a comparison between the proposed safety-enhanced diffuser method with the safe RL methods.
4. For the Primal Dual and ALM methods there are chances that the control will still violate the constraints even in the simpler time invariant cases. It would be nice if the authors could do more worst-case analysis and discuss how bad the performance will be in extreme cases to help the reader evaluate the risk of the method.

---

> ### Author Rebuttal · Authors · 2025-07-31
>
> We thank the reviewer for all the constructive comments. We are glad the reviewer found our work to be an "interesting and important topic." We will now address each of your comments in detail.
>
> **Q1: The performances of SafeDiffuser in time-varying scenarios are not shown.**
>
> **Response:** Thank you for this observation. The problem comes from a mismatch between states and actions in the original SafeDiffuser when moving from planning to online execution. SafeDiffuser uses CBF during reverse diffusion to create a safe state trajectory in planning. However, the actions used in execution are not based on this planned safe state. The model is trained on both states and actions, but during diffusion, only the states are guided, and actions are sampled directly without adjustment. This leads to a mismatch where the executed actions may not follow the planned safe states. Therefore, the original method is mainly suitable for planning, and online execution results were not reported. To compare fairly, we ran additional experiments combining SafeDiffuser with our IDM on all tasks. The results, shown in the table, show SafeDiffuser works well in planning but performs less well online. We will include these results in the revised paper.
> |Algorithm|Env|Reward|PlanningViolations|Impl.Violations|ViolationRates(%)|Time/s|
> |-|-|-|-|-|-|-|
> |SafeDiffuser(IDM)|Hopper|3514±46|0.00±0.00|0.28±0.06|0.27±0.05|0.9307|
> ||Swimmer|60.4±6.1|0.00±0.00|0.65±0.21|0.97±0.25|0.9301|
> ||SafetyBallRunning|820.8±29.8|--|1.13±1.08|21.9±12.8|2.6e-2|
>
>
> **Q2: How to guarantee the consistency of the controls due to the significance in practice.**
>
> **Response:** In some dynamical systems, an inverse dynamical model can indeed produce multiple control solutions for a given state transition. The mapping from desired state transitions to control inputs is not always one-to-one, especially in systems with redundancy or underdetermined dynamics. This is a known challenge and has been studied extensively in both robotics and aerospace communities. One effective approach is to introduce additional constraints or optimization criteria. For example, if multiple controls can achieve the same state transition, selecting the one that minimizes a cost function -- such as energy consumption, control effort, or time -- can provide a unique and consistent solution. In our research, we use a deterministic neural network to provide the action output which is unique when we make the assumption that the inverse dynamics models are available. To avoid  inconsistency, we could always add a regularization term to the loss function during training to explicitly encourage minimum-effort solutions that may generalize well within the training distribution. Moreover, the inverse dynamics model is queried on well-behaved transitions whose distribution closely matches the expert distribution, which further reduces the risk of inconsistency. In future work, this framework could be enhanced to more explicitly handle ambiguity in complex or underactuated systems.
>
> **Q3: Summarize the constrained sampling works for safe planning and control like safe RL and draw a comparison.**
>
> **Response:** We thank the reviewer for raising this important point. While constrained sampling has been explored in safe RL, our work differs significantly in both the problem formulation and the solution paradigm. For Problem Setting, previous safe RL methods typically formulate the problem either as a Constrained Markov Decision Process (CMDP)[1][2], focusing on expected cumulative costs and enforcing trajectory-level constraints, or impose state-wise constraints via CBF[3] or reachability analysis[4]. Beyond safe RL, several other methods have also been proposed for constrained sampling. Task-constrained RRT [5] employs tangent spaces for manipulators under general constraints while they usually have the shortcoming that computing the kernel of the constraint Jacobian is computationally expensive and may require multiple matrix decompositions. These methods are generally complex to implement, which prevents widespread applicability to many different robotic systems[6].
>
> Our work explicitly considers state-wise constraints during planning and control, ensuring that individual states along the trajectory remain within safety bounds. This is particularly important in safety-critical applications where violations at any step (not just on average, as is commonly done in many prior safe RL studies) can be unacceptable. For Solution Approach, previous safe RL methods usually aim to learn, from scratch, a policy by jointly optimizing performance and satisfying constraints. This usually leads to a sophisticated learning process and requires a large number of interactions with the environment. Alternatively, some approaches solve complex constrained optimization problems during training by combining the RL controller with safety filters (typically QP-based). However, such approaches can be computationally demanding and may impose additional requirements on the system dynamics, such as the need for a control-affine structure to ensure tractable optimization. In contrast, our method follows a modular two-stage framework:
>
> 1. We first train a powerful trajectory generator (diffusion model) to model expert behaviours without enforcing safety constraints.
>
> 2. Then, we inject safety at sampling time via constraint-guided denoising, producing trajectories that satisfy state-wise constraints.
>
> The IDM can be then applied to generate safe actions based on the planned safe trajectories. This decoupling offers several advantages:
>
> 1. Modularity: Safety and performance are handled independently, allowing for better reuse and flexibility.
>
> 2. No specific requirements on the dynamics model: Unlike many QP-based filtering methods in prior safe RL studies, our approach does not require specific structural assumptions such as control-affine dynamics.
>
> 3. Lower computational burden: The diffusion model is trained offline, and constraint satisfaction is imposed only during sampling, avoiding the need to backpropagate safety constraints throughout training.
>
> 4. Expressivity: While RL policies are typically modeled as multivariate Gaussian distributions, diffusion models can represent complex and multimodal trajectory distributions effectively.
>
> We will add a discussion in the revised manuscript to clarify these distinctions and summarize related works. We believe our framework offers a scalable and flexible alternative to traditional safe RL.
>
> [1] Achiam, J., Held, D., Tamar, A. and Abbeel, P., 2017, July. Constrained policy optimization. In International conference on machine learning (pp. 22-31). PMLR.
>
> [2] Tessler, C., Mankowitz, D.J. and Mannor, S., 2018. Reward constrained policy optimization. arXiv preprint arXiv:1805.11074.
>
> [3] Cheng, R., Orosz, G., Murray, R.M. and Burdick, J.W., 2019, July. End-to-end safe reinforcement learning through barrier functions for safety-critical continuous control tasks. In Proceedings of the AAAI conference on artificial intelligence (Vol. 33, No. 01, pp. 3387-3395).
>
> [4] Yu, D., Ma, H., Li, S. and Chen, J., 2022, June. Reachability constrained reinforcement learning. In International conference on machine learning (pp. 25636-25655). PMLR.
>
> [5] Stilman, M., 2010. Global manipulation planning in robot joint space with task constraints. IEEE Transactions on Robotics, 26(3), pp.576-584.
>
> [6] Kingston, Z., Moll, M. and Kavraki, L.E., 2018. Sampling-based methods for motion planning with constraints. Annual review of control, robotics, and autonomous systems, 1(1), pp.159-185.
>
> **Q4: Do worst-case analysis and discuss how bad the performance will be in extreme cases to help evaluate the risk of the method**
>
> **Response:** Our paper provides a quantitative analysis of the trade-offs in its experimental tables, which show the mean and standard error of constraint violations across various tasks, offering insight into the methods' typical performance and variance. For practical deployment, one could adopt more robust strategies, such as defining more conservative constraints, to ensure safe outcomes even in worst-case scenarios.
> To more directly address your concern, we have conducted an additional worst-case analysis experiment in the PyBullet Safety BallRun environment. In the original setup, the agent and the moving obstacle start with a safe distance between them. In this new analysis, we created more extreme initial conditions by placing the obstacle at different starting positions to our agent:
>
> Worst-Case: Starting at the point of collision (zero margin).
>
> |Env|SafetyMargin|Algorithm|Reward|Impl.Violations|ViolationRates(%)|
> |-|-|-|-|-|-|
> |SafetyBallRunning|Boundary|Diffuser|640.6±29.0|14.04±1.27|81.9±1.92|
> |||PrimalDual|457.1±48.4|2.99±0.12|67.5±2.3|
> |||ALM|380.9±27.1|3.10±0.17|70.1±1.5|
> ||10%|Diffuser|640.8±28.9|13.08±1.40|76.5±2.2|
> |||PrimalDual|479.5±54.6|1.55±0.23|32.5±5.6|
> |||ALM|421.9±59.77|1.33±0.21|29.0±4.4|
> ||25%|Diffuser|640.6±29.0|11.42±1.65|69.2±2.99|
> |||PrimalDual|413.9±87.9|0.45±0.75|18.9±16.5|
> |||ALM|422.1±70.4|0.06±0.11|8.0±8.5|
>
> The results demonstrate the robustness of our approach. Even in the most extreme case starting at the point of collision, our methods still effectively reduce the magnitude and frequency of constraint violations compared to the baseline. As the initial safety margin increases to 10% and 25%, the number of violations is further and substantially reduced, showcasing the framework's ability to maintain safety even when initialized in or very near to a compromised state.

---

> > ### Comment · Reviewer_SjPT · 2025-08-07
> > **Question about Worst-Case Analysis**
> >
> > > Worst-Case: Starting at the point of collision (zero margin).
> >
> > The reviewer appreciates the addition of the worst-case experiments. One curiosity: Why did you not evaluate the performance of the "Projected" method here?

---

> > > ### Author Response · Authors · 2025-08-08
> > > **Response to Reviewer SjPT's comment**
> > >
> > > We apologize for the confusion and thank you for pointing out this omission. You are correct; our worst-case analysis should have included the Projected method. This was an oversight in our initial rebuttal preparation, where we mistakenly omitted its results.
> > >
> > > We have now added the data for the Projected method to provide a complete picture and will add this part in the revised paper.
> > >
> > > |Env|SafetyMargin|Algorithm|Reward|Impl.Violations|ViolationRates(%)|
> > > |-|-|-|-|-|-|
> > > |SafetyBallRunning|Boundary|Diffuser|640.6±29.0|14.04±1.27|81.9±1.92|
> > > |||Projected|536.7±36.2|3.44±0.32|74.6±1.6|
> > > |||PrimalDual|457.1±48.4|2.99±0.12|67.5±2.3|
> > > |||ALM|380.9±27.1|3.10±0.17|70.1±1.5|
> > > ||10%|Diffuser|640.8±28.9|13.08±1.40|76.5±2.2|
> > > |||Projected|507.1±57.6|2.63±0.36|57.8±2.0|
> > > |||PrimalDual|479.5±54.6|1.55±0.23|32.5±5.6|
> > > |||ALM|421.9±59.77|1.33±0.21|29.0±4.4|
> > > ||25%|Diffuser|640.6±29.0|11.42±1.65|69.2±2.99|
> > > |||Projected|525.0±59.2|0.31±0.16|19.6±3.7|
> > > |||PrimalDual|413.9±87.9|0.45±0.75|18.9±16.5|
> > > |||ALM|422.1±70.4|0.06±0.11|8.0±8.5|

---

> ### Comment · Reviewer_3UpD · 2025-08-04
>
> Thank you for the authors' replies. Most of my concerns have been resolved. I have one more question. The authors mentioned that
> >  In our research, we use a deterministic neural network to provide the action output which is unique when we make the assumption that the inverse dynamics models are available.
>
> Could you elaborate on this a bit more? Especially about the choice of NN used here and why an alternative classical choice (like solve for the action from the modeled dynamics) is not used.

---

> > ### Author Response · Authors · 2025-08-04
> > **Response to Reviewer’s Comment**
> >
> > Thank you for the valuable follow-up question.
> >
> > In our work, we choose a deterministic inverse dynamics model implemented with multilayer perceptrons. We discretize the action space into a finite number of bins and treats the problem as a sequence of classification tasks. The model first encodes the concatenated initial and next states into a latent feature vector. Then, it generates the action vector autoregressively, dimension by dimension where each dimension is predicted using the multilayer perceptron that takes both the state feature vector and an embedding of all previously generated action dimensions as input. **The autoregressive structure captures dependencies between action dimensions to generate more coordinated actions, which is different from traditional neural networks.** Additionally, by discretizing the action space, it effectively transforms a regression problem into a more stable classification problem. The implementation follows [1]. In future work, we will also explore more expressive and efficient IDM architectures, such as Neural ODE–based models, to better capture complex continuous-time dynamics.
> >
> > We would like to clarify that we assume **the inverse dynamics models exist but are not directly accessible in closed form from first-principle physics**. The forward dynamics are black-box during trajectory generation, making classical model-based inversion infeasible. Instead, we train this deterministic neural IDM offline using data collected from interactions with the true dynamics, so that the learned neural network based model is “available” during deployment and can convert safe trajectories into actions efficiently with a single neural-network forward process. This design ensures both state–action consistency and high computational efficiency in real deployment. We will add this clarification in the revised paper.
> >
> > Please feel free to reach out if there are any further questions.
> >
> > [1] Ajay, Anurag, et al. "Is conditional generative modeling all you need for decision-making?." arXiv preprint arXiv:2211.15657 (2022).

---

> > > ### Comment · Reviewer_3UpD · 2025-08-04
> > >
> > > Thank you for the quick answer. I have no more questions and will adjust my score accordingly. Besides, I suggest that this content about the inverse dynamic model should be added to the paper to prevent misunderstanding.

---

> > > > ### Author Response · Authors · 2025-08-05
> > > > **Response to Reviewer’s Comment**
> > > >
> > > > Thank you for your positive feedback. We are glad that our response addressed your concerns and appreciate your willingness to adjust your score. We also value your suggestion to include more details about the inverse dynamic model, and we will integrate this in the revised paper to ensure greater clarity.
> > > >
> > > > Thank you again for your time and for helping us improve our work.

---

> ### Comment · Reviewer_SjPT · 2025-08-07
> **Disadvantages of the IDM approach (e.g., in underactuated settings)**
>
> > The IDM can be then applied to generate safe actions based on the planned safe trajectories. This decoupling offers several advantages:
> > - Modularity: Safety and performance are handled independently, allowing for better reuse and flexibility.
> > - No specific requirements on the dynamics model: Unlike many QP-based filtering methods in prior safe RL studies, our approach does not require specific structural assumptions such as control-affine dynamics.
> > - Lower computational burden: The diffusion model is trained offline, and constraint satisfaction is imposed only during sampling, avoiding the need to backpropagate safety constraints throughout training.
> > - Expressivity: While RL policies are typically modeled as multivariate Gaussian distributions, diffusion models can represent complex and multimodal trajectory distributions effectively.
>
> The reviewer appreciates the author's explanation and generally agrees with these points. However, for a fair comparison, the authors should also make sure to point out the disadvantages and limitations of such an approach in the revised paper. Specifically, I am thinking here of two scenarios:
>
> - For underactuated robots, many states are not attainable - especially not at steady state. Therefore, an action might not even exist that is able to drive the system towards the planned next state.
> - Similarly, actuation bounds and constraints might render a planned state transition infeasible, because, for example, the actuation cannot accelerate the robot sufficiently fast. Do you account for such scenarios in your experiments? Would your method be able to deal with such constraints?

---

> > ### Author Response · Authors · 2025-08-08
> > **Response to Reviewer SjPT's comment II**
> >
> > Thank you for raising these critical and practical points.
> >
> > In our framework, both the diffusion model and the Inverse Dynamics Model (IDM) learn from expert trajectories. These trajectories are gathered from interaction with the physical system and inherently contain only dynamically feasible state-action sequences. Therefore, with the assumption of sufficient training steps, the diffusion model learns the distribution of these feasible trajectories, and the IDM learns the corresponding valid inverse dynamics.
> >
> > However, you have correctly identified a key challenge: for underactuated systems, the planner could generalize outside the expert data distribution and propose a kinematically impossible transition. We believe our proposed framework itself offers a path to mitigate this. Our method for constrained trajectory generation is general. While the experiments in this paper focused on state-based constraints to demonstrate the core capabilities, the framework can be extended to handle constraints related to dynamic feasibility and actuation limits. Specifically for actuation constraints, we can generate both state and action trajectories, and define a constraint on the output of the IDM, such as $IDM(x^\tau, x^{\tau+1}) =u^\tau, u^\tau \leq u_{\max}$. These new constraints can be enforced during the reverse diffusion process alongside other safety constraints, ensuring that the planner only generates trajectories that are executable within the given actuation limits and that all planned states are reachable under the ideal assumption that the diffusion model and IDM are accurate.
> >
> > In this paper, our primary goal was to introduce the constrained diffuser framework and validate its effectiveness. For this reason, we chose to focus our experiments on state-based constraints. We are very grateful for you pointing out these important limitations. Ensuring the existence of control action for underactuated robots is an important topic but slightly different from this paper's scope and we plan to leave it for future work. We will add a thorough discussion of these points and the potential extensions of our framework in the revised manuscript.

---

> > > ### Comment · Reviewer_SjPT · 2025-08-08
> > >
> > > > We apologize for the confusion and thank you for pointing out this omission. You are correct; our worst-case analysis should have included the Projected method. This was an oversight in our initial rebuttal preparation, where we mistakenly omitted its results.
> > > > We have now added the data for the Projected method to provide a complete picture and will add this part in the revised paper.
> > >
> > > Thank you for adding the results of the "Projected method".
> > >
> > > > In our framework, both the diffusion model and the Inverse Dynamics Model (IDM) learn from expert trajectories. These trajectories are gathered from interaction with the physical system and inherently contain only dynamically feasible state-action sequences. Therefore, with the assumption of sufficient training steps, the diffusion model learns the distribution of these feasible trajectories, and the IDM learns the corresponding valid inverse dynamics.
> > > > ..
> > > > In this paper, our primary goal was to introduce the constrained diffuser framework and validate its effectiveness. For this reason, we chose to focus our experiments on state-based constraints. We are very grateful for you pointing out these important limitations. Ensuring the existence of control action for underactuated robots is an important topic but slightly different from this paper's scope and we plan to leave it for future work. We will add a thorough discussion of these points and the potential extensions of our framework in the revised manuscript.
> > >
> > > Thank you for this explanation and the reviewer agrees with this perspective. It would seem valuable to point out this interesting future research direction in the paper and show a pathway how such actuation/underactuation constraints could be incorporated into the framework.

---

> > > > ### Author Response · Authors · 2025-08-09
> > > > **Response to Reviewer SjPT's comment**
> > > >
> > > > Thank you for your positive feedback and for this excellent suggestion.
> > > >
> > > > In the revised manuscript, we will be sure to add this as a future research direction in our discussion. As you suggested, we will not only mention it but also briefly outline a potential pathway for how such constraints could be incorporated into our framework.
> > > >
> > > > We appreciate your insightful feedback in helping us improve the paper.

---

### Official Review · Reviewer_x55D · 2025-07-02

**Clarity:** 4
**Significance:** 3
**Originality:** 3
**Rating:** 5
**Confidence:** 3

**Summary:**

This paper proposes Constrained Diffusers, a method for enforcing safety constraints in planning and control using pretrained diffusion models. Rather than modifying the model or retraining, the authors modify the reverse sampling process using constrained Langevin dynamics with three algorithmic variants: Projected Gradient, Primal-Dual, and Augmented Lagrangian. The framework supports discrete control barrier functions (DCBFs) to enforce safety online. Experiments in Maze2D, MuJoCo locomotion, and PyBullet demonstrate the efficacy and efficiency of these methods under static and time-varying constraints.

**Questions:**

- Can you compare the three methods? What are the pros and cons of each one?

- If I understand correctly, while (9) is formulated as a general constraint, later on the constraint is chosen as the constraint barrier function. I wonder what’s the difference between using CBF as constraint vs. directly impose safety set constraints on the states? Further, as the proposed method generates state trajectory, how to ensure the state trajectory is dynamically feasible?

- In Theorem 1, does a local saddle point always satisfy constraint?

**Ethical Concerns:**

["NO or VERY MINOR ethics concerns only"]

**Final Justification:**

I appreciate the reviewer for the detailed response. The comparison of the three is nice. I keep my (positive) score as is.

**Limitations:**

Yes

**Quality:**

4

**Strengths And Weaknesses:**

Strength:

- The paper addresses a critical gap in diffusion-based planners: enforcing constraints without retraining. This is highly relevant for real-world safety-critical systems.

- This paper has nice connection to classical constrained optimization. By formulating constraint satisfaction as KL-regularized sampling, the authors connect reverse diffusion with well-established constrained optimization principles.


Weakness:
I do not see major weakness of the paper. See questions.

---

> ### Author Rebuttal · Authors · 2025-07-31
>
> We thank the reviewer for their positive and insightful feedback on our work. We are pleased that they recognized the key strengths of our paper, and we will now address the specific questions raised in detail.
>
> **Q1: Pros and Cons of the three methods**
>
> **Response: Projected Method** :This method modifies the standard diffusion update rule by projecting the generated sample onto the feasible set defined by the constraints at each denoising step. **Pros:**- This method provides the strictest safety guarantee among the three methods. It ensures constraints are met at every diffusion step and consistently achieves zero planning violations in experiments. **Cons:** The projection operation can be time-consuming especially when constraints are complex because we need to solve an optimization problem, making this method significantly slower than the Primal-Dual and ALM approaches.
> **Primal-Dual (PD) Method**: This approach treats the problem as a constrained optimization task, using Lagrange multipliers to incorporate constraints. It simultaneously updates the trajectory (primal variable) and multipliers (dual variables) using gradient-based steps. **Pros:** This method has a high computational efficiency with computation times often comparable to the unconstrained baseline diffuser and admitstheoretical convergence analysis even for nonconvex constraints based on the assumptions given in the paper.  **Cons:** This method does not may not practically guarantee strict constraint satisfaction in some experiments because more diffusion steps are needed for convergence. The final generated trajectories often have small, non-zero violations.
> **Augmented Lagrangian Method (ALM)** ALM enhances the Primal-Dual method with another quadratic penalty term to the Lagrangian function. **Pros:**: This method has the best balance of computational efficiency and safety:  Similar to the PD method, it slightly increases the computational time and provides better constraint satisfaction compared to the standard Primal-Dual method. **Cons:**  Like the PD method, it does not offer the absolute guarantee of constraint satisfaction of the Projected method. The convergence analysis is based on some strong assumptions which may be hard to guarantee in real-world experiments.
>
> **Q2: Difference between using CBF as constraint vs. directly impose safety set constraints on the states**
>
> **Response:** The reviewer is correct that the paper first formulates a general constraint problem and then uses a Discrete Control Barrier Function (DCBF) as a specific type of constraint. Safety Set Constraint is a static constraint applied to each individual state in a trajectory. If the safe set is defined as $\\mathcal{C} = \\{x \in \\mathbb{R}^d : h(x) \\ge 0\\}$, a direct constraint would simply enforce that for every state $x^\\tau$ in the planned trajectory, the condition $h(x^\\tau) \\ge 0$ holds. This ensures each point of the plan is in a safe location. The obstacle avoidance constraints in the Maze2D experiments are an example of this. The Discrete Control Barrier Function (DCBF) constraint provides a stronger, dynamics-aware safety guarantee due to input constraints. Instead of just ensuring each state in the safe set, it imposes a condition on the transition between consecutive states to ensure that the safety margin does not decrease too quickly. The goal of a DCBF is to guarantee the forward invariance of the safe set, meaning that if the system starts in a safe state, the entire trajectory generated under this constraint will remain safe, ensuring the trajectories plan from one point to the next is safe. Also, the dynamic feasibility of the generated state trajectory is ensured by using a learned Inverse Dynamics Model (IDM) combined with DCBF. After the constrained diffuser generates a safe sequence of states $(x^0,x^1,...,x^\mathcal{T})$, the framework uses the IDM to determine the action required for each transition and is then fed into the environment. This approach relies on the assumption that an IDM exists for these tasks. Since the expert trajectories are by definition dynamically feasible, the IDM learns the underlying dynamics of the system. DCBF then plays a role in providing safe guarantees on the actions from the IDM, ensuring that the generated safe state trajectory can be translated into a safe actions. We give an experimental comparison below. The reviewer can also refer to Section G in Appendix for more details.
> | Env                | Algorithm     | Planned Violations | Planned Violations (w/o CBF) | Implementing Violations | Implementing Violations (w/o CBF) |
> |--------------------|---------------|-------------------|-----------------------------|------------------------|----------------------------------|
> | **Hopper**         | Projected     | **0.00 ± 0.00**   | **0.00 ± 0.00**             | **0.29 ± 0.09**        | 0.42 ± 0.10                     |
> |                    | Primal Dual   | 0.10 ± 0.03       | 0.13 ± 0.03                 | 0.43 ± 0.12            | 0.46 ± 0.43                     |
> |                    | ALM           | 0.08 ± 0.02       | 0.10 ± 0.03                 | 0.38 ± 0.16            | 0.61 ± 0.12                     |
> | **Swimmer**        | Projected     | **0.00 ± 0.00**   | **0.00 ± 0.00**             | 0.61 ± 0.25            | 0.75 ± 0.29                     |
> |                    | Primal Dual   | 0.17 ± 0.08       | 0.16 ± 0.06                 | 0.02 ± 0.02            | 0.06 ± 0.08                     |
> |                    | ALM           | 0.08 ± 0.05       | 1.80 ± 0.52                 | **0.00 ± 0.02**        | 0.42 ± 0.24                     |
> | **SafetyBallRunning** | Projected  | -                 | -                           | 1.01 ± 0.63            | 1.50 ± 0.14                     |
> |                    | Primal Dual   | -                 | -                           | 0.02 ± 0.03            | 0.10 ± 0.13                     |
> |                    | ALM           | -                 | -                           | **0.02 ± 0.02**        | **0.02 ± 0.02**                 |
>
> **Q3: Does a local saddle point always satisfy constraint?**
>
> **Response:** Yes. According to Definition 4.1 in the paper, a point $(q*, \\lambda*)$ is a local saddle point if and only if it satisfies, for all $\\lambda \\ge 0$: $L(q*, \\lambda*) \\ge L(q*, \\lambda).$  We have the following Lagrangian function as: $L(q, \\lambda) = KL(q \\| p) + \\lambda^\\top \\mathbb{E}_{x \\sim q}[g(x)].$  Then we obtain $KL(q* \\| p) + (\\lambda*)^\\top \\mathbb{E}[g(x*)] \\ge KL(q* \\| p) + \\lambda^\\top \\mathbb{E}[g(x*)].$ Simplifying gives  $(\\lambda* - \\lambda)^\\top \\mathbb{E}[g(x*)] \\ge 0.$  This inequality must hold for all non-negative $\\lambda$ (i.e., $\\lambda \\ge 0$). Suppose that at the saddle point $(q*, \\lambda*)$, the constraint is not satisfied. This implies that at least one constraint is positive, i.e., some component in $\\mathbb{E}[g(x*)]$ is greater than $0$.  Under this assumption, we check whether the inequality  $(\\lambda^* - \\lambda)^\\top \\mathbb{E}[g(x*)] \ge 0$  can still hold for all $\\lambda \\ge 0$. Choose a very large positive value $\lambda$, in the dimensions where $\\mathbb{E}[g(x*)] > 0$. Then $(\\lambda* - \\lambda)$ becomes highly negative in those dimensions. Consequently, the inner product  $(\lambda^* - \\lambda)^\\top \\mathbb{E}[g(x*)]$  becomes a negative number which violates the inequality  $(\\lambda* - \\lambda)^\\top \\mathbb{E}[g(x*)] \\ge 0,$  creating a direct contradiction. Therefore, the saddle point condition must also satisfy the constraint, i.e.,  $\\mathbb{E}[g(x*)] \\le 0.$  Please refer to Section E, Step 3 in the Appendix from supplementary materials for a detailed proof.

---

### Official Review · Reviewer_UkXA · 2025-07-05

**Clarity:** 3
**Significance:** 2
**Originality:** 2
**Rating:** 3
**Confidence:** 5

**Summary:**

The paper introduces Constraint Diffusers,  a framework that incorporates constraints into pre-trained diffusion models without retraining or architectural modifications, by incorporating constrained barrier functions into the model. The objective is to make diffusion predictions safe and use them with planning. Constrained Diffusers aim to inject safety constraints into pretrained diffusion-based models that generate full action sequences or trajectories. The method recognizes that retraining large diffusion models to account for new constraints is computationally expensive. Instead, they modify the sampling process to ensure constraint satisfaction post hoc.

**Questions:**

How does your work compare with current literature that uses barrier functions to enhance the safety of planning systems?

Why is this method useful when in fact there are closed form model-based and computationally efficient solutions for the problem you address? How does the method compare to other planning methods like MPC with safety layers or goal-conditioned RL with constraints?

How sensitive is constraint enforcement to the quality of the inverse dynamics model? What happens when the IDM is inaccurate or poorly generalized?

Do these constrained sampling methods scale well to high-dimensional action spaces (e.g., humanoid robotics)?

How does the method perform when multiple competing constraints (e.g., safety vs. performance) must be balanced?

Is there a quantitative analysis of trade-offs between computation time and constraint satisfaction for each method?

What assumptions are made about access to constraints, and how realistic are those assumptions in practical deployment?

**Ethical Concerns:**

["Major Concern: Data quality and representativeness"]

**Limitations:**

No this section is missing from the paper

**Paper Formatting Concerns:**

No concerns

**Quality:**

3

**Strengths And Weaknesses:**

Pros:
The method works directly with pretrained diffusion models, avoiding expensive retraining.
The method makes safety a post-training concern, improving modularity and practical usability.
The paper offers a toolkit of constraint-handling strategies

Cons:
Success in enforcing complex, nonconvex, or time-varying constraints is dependent on the optimization dynamics—true safety guarantees may still be fragile.
Experiments are simple and preliminary. Experiments are done in simulation only on toy examples. The results look very preliminary.
No mention of prior control barrier function methods. How is your paper improving on/ different from several papers on the same topic which are not cited, specifically mapaers that use CBFs with ML such as BarrierNet and papers that address safe diffusion such as Safediffuser: Safe planning with diffusion probabilistic models (ICML 2025 and arXiv)?
The solution requires accurate learning or specification of inverse dynamics to translate constrained trajectories into executable actions.

---

> ### Author Rebuttal · Authors · 2025-07-31
>
> We thank the reviewer for the constructive review and feedback. We have carefully addressed each of your points below.
>
> **W1: Fragile Guarantees**
>
> **Response:** The success of PD and ALM methods depends on the optimization dynamics. Multi-scale analysis of PD updates uses two learning rates and assumes a local optimum near the initialization, without requiring global convexity. The same applies to ALM, which drives constraint violations asymptotically to zero with sufficient iterations. Our work also offers a trade-off by providing different kinds of solutions: For hard safety guarantees, the Projected method enforces strict constraints. For efficiency, PD and ALM offer practical alternatives by replacing expensive optimization with simple updates.
>
> **W2: Preliminary Experiment**
>
> **Response:** Our goal is to propose a foundational framework for diffusion models that addresses critical safety challenges and offers a spectrum of solutions. To this end, we evaluate it in standard, well-understood environments such as MuJoCo and PyBullet to demonstrate effectiveness under different safety requirements and validate its broad applicability. Following your suggestion, we also conducted additional experiments on a high-dimensional dexterous hand manipulation task, detailed in our response to Q4.
>
> **W3: Prior Work**
>
> **Response:** We extensively discuss related work, including SafeDiffuser, in Section 7 Related Work and compare to it in our experiments. [1] proposes BarrierNet, a differentiable layer that embeds learnable CBF parameters into neural networks for end-to-end training of less conservative policies. Other studies address safety in imitation learning and uncertainties through solve QP [2]-[4]. Compared to SafeDiffuser, our key difference is that it applies CBFs at intermediate diffusion steps, whereas we enforce CBFs along the entire trajectory in the physical world. Moreover, most prior CBF methods rely on solving costly QPs, while two of our methods exploit diffusion properties to impose CBFs efficiently without per-step constrained optimization.
>
> **W4: Accurate Inverse Model**
>
> **Response:** We agree that the accuracy of the IDM is crucial for converting planned trajectories into actions. Our experiments show the learned IDM performs well on the tasks studied. To further investigate, we conducted additional tests with an inaccurate IDM, detailed in our response to Q3. Future work will focus on improving robustness by exploring advanced models like PINNs and Neural ODEs, and designing policies less sensitive to IDM errors.
>
> **Q1: Compare to other barrier function literatures**
>
> **Response:** Compared to the current literature discussed above, we use diffusion models as planner due to their exceptional ability to represent complex and multimodal distributions over trajectories. It provides advantage over traditional planning methods that may have difficulty capturing the diversity of expert behaviours. While the current literature has successfully combined CBF for safe planning, our work introduces a different paradigm that leverages the properties of diffusion processes with gradient-based techniques to iteratively guide the diffusion process toward satisfying the safety constraints, leading to a significant reduction in computation time.
>
> **Q2: Why methods useful? Compare to other planning methods?**
>
> **Response:** Our method is suited for scenarios where the objective is unknown and policies are learned from expert demonstrations, where model-based MPC is often impractical. Constraints may allow a closed-form solution, but integrating them into behavioral cloning is challenging. Our Primal-Dual and ALM approaches address this by only requiring the gradient of each constraint. Compared to learning-based MPC with safety layers or goal-conditioned RL with constraints, our diffusion framework has several advantages. Diffusion models naturally learn multimodal distributions from expert data, generating diverse trajectories that adapt well to changing environments. Our gradient-based constraint enforcement is also more efficient than typical QP-based safety layers. Diffusion models also support vision and language inputs, enabling our framework to scale to more complex, real-world problems than many traditional planning algorithms.
>
> **Q3: How sensitive is constraint to IDM? What happens with inaccurate IDM?**
>
> **Response:** The accuracy of the IDM is critical because errors cause a mismatch between planned and actual states, leading to potential task failures and safety violations. We quantify this sensitivity by defining the IDM error as $\\Delta u = \\tilde{u}^\\tau - u^\\tau$. Linearizing the dynamics shows the resulting state deviation $\\Delta x \\approx \\frac{\\partial f_d}{\\partial u} \\Delta u$, which affects the safety function as $h(x_{actual}^{t+1}) \\approx h(x_{plan}^{t+1}) + \\nabla h(x_{plan}^{t+1})^\\top \\Delta x,$ meaning the change in constraints is primarily influenced by $\\nabla h(x^{t+1}_{plan})^\\top \\frac{\\partial f_d}{\\partial u}$. We conducted sensitivity experiments on Hopper with varying IDM errors (2%, 10%, 25%) and evaluated three algorithms (Projected, Primal-Dual, ALM). Results show that increasing IDM error generally lowers reward and increases constraint violations, but a 10% error is still acceptable for the tasks (see table below).
>
> |Env|Error|Algorithm|Reward|Impl.Violations|ViolationRate(%)|
> |-|-|-|-|-|-|
> |Hopper|2%|Projected|3437±196|0.28±0.12|1.76±0.75|
> |||PrimalDual|3501±130|0.48±0.214|3.47±0.85|
> |||ALM|3370±442|0.44±0.19|3.13±0.10|
> ||10%|Projected|3391±325|0.32±0.08|1.98±0.60|
> |||PrimalDual|2690±707|0.46±0.12|3.33±0.86|
> |||ALM|2369±455|0.41±0.08|3.09±0.02|
> ||25%|Projected|1900±471|0.62±0.12|2.09±0.29|
> |||PrimalDual|1446±354|1.19±0.22|4.67±0.75|
> |||ALM|1614±504|0.95±0.33|4.11±0.02|
>
> Additional swimmer experiments will be included in the revised paper due to space constraints.
>
> **Q4: Scalibility to high-dimensional action spaces**
>
> **Response:** Compared to traditional algorithms that suffer from the curse of dimensionality, learning-based methods handle high-dimensional action spaces better through end-to-end learning and strong nonlinear fitting, without explicit search. We conducted additional experiments on the 30-dimensional Dexterous Hand Manipulation Adroit Relocate task from D4RL with two translation constraints. Results summarized below (details to appear in the revised paper) show that even with similarly sized models, our algorithms satisfy constraints, demonstrating scalability to high-dimensional tasks.
>
> |Env|Algorithm|Rewards|Violations|ViolationRates(%)|Time(s)|
> |-|-|-|-|-|-|
> |Relocation|Diffuser|550±938|87.0±30.4|74.0±16.4|0.0028|
> ||Conditional|422±837|32.0±14.7|58.5±16.3|0.0029|
> ||Projected|321±791|0.0±0.0|0.0±0.0|0.9515|
> ||PrimalDual|499±951|0.04±0.03|2.9±1.9|0.0031|
> ||ALM|325±670|0.04±0.02|2.9±1.6|0.0035|
>
> **Q5: Performance when multiple competing constraints must be balanced?**
>
> **Response:** Our method formulates diffusion as a constrained optimization problem with safety as the top priority. The algorithm first finds feasible solutions satisfying all constraints, then searches along the safe boundary under some assumptions, keeping safety dominant even when optimizing performance. From a dual perspective, each constraint has a Lagrange multiplier updated during reverse diffusion to push violated trajectories back into the feasible region. When constraints compete, the process behaves like a dynamic game, minimizing the original objective while balancing a weighted sum of violations to explore the Pareto frontier. We often relax non-critical constraints by adding them to the objective function or introduce slack variables to maintain overall feasibility. We enforce Control Barrier Functions (CBFs) over entire trajectories, which guarantees safety but risks local trapping; stochastic denoising mitigates this by occasionally escaping traps, balancing strict safety with feasible performance exploration.
>
> **Q6: Quantitative analysis of trade-offs?**
>
> **Response:** Based on the experimental data, we analyze the trade-off between computation time and constraint satisfaction. The Projected Method achieves zero violations but incurs high computation cost due to solving an optimization at every diffusion step. Primal-Dual is highly efficient, running near the speed of vanilla diffusion with only minor violations. ALM strikes a balance, trading slightly more computation for improved constraint satisfaction. A more detailed analysis will be added in the revised paper.
>
> **Q7: Assumptions made on constraints, and how realistic in practical deployment?**
>
> **Response:** Our framework assumes a differentiable constraint function and its gradient. For common cases like geometric obstacles, this can be derived analytically. In more complex settings, constraints can be built in real time from sensor data, e.g., camera-based segmentation for obstacle boundaries. For black-box constraints, a neural network can learn a differentiable approximation from cost signals collected via environmental interactions.
>
> [1] Xiao, W., et al. (2023). "BarrierNet: Differentiable Control Barrier Functions for Learning of Safe Robot Control." IEEE Transactions on Robotics, vol. 39, no. 3, pp. 2289-2307, June 2023, doi: 10.1109/TRO.2023.3249564.
>
> [2] Cosner, R. K., Yue, Y., and Ames, A. D. (2022). "End-to-End Imitation Learning with Safety Guarantees using Control Barrier Functions." 2022 IEEE 61st Conference on Decision and Control (CDC).
>
> [3] W. Xiao, C. Belta and C. G. Cassandras, "Adaptive Control Barrier Functions," in IEEE Transactions on Automatic Control, vol. 67, no. 5, pp. 2267-2281, doi: 10.1109/TAC.2021.3074895.
>
> [4] Taylor, A. J., Singletary, A., Yue, Y., and Ames, A. D. (2020). "Learning for Safety-Critical Control with Control Barrier Functions." 2nd Annual Conference on Learning for Dynamics and Control.

---

> > ### Comment · Reviewer_UkXA · 2025-08-02
> >
> > Thank you for the responses. What data sets did you use for the new experiments?

---

> > > ### Author Response · Authors · 2025-08-02
> > > **Response to Reviewer’s Question**
> > >
> > > Thank you for your question. The new environment Adroit Relocate was introduced in [5] as part of the Adroit manipulation platform. For our new experiments, we use the **D4RL dataset relocate-expert-v1**, which contains 5,000 expert trajectories sampled from an expert that solves the task.
> > >
> > > Please feel free to let us know if you have any additional questions.
> > >
> > > [5] Rajeswaran, A., et al. Learning Complex Dexterous Manipulation with Deep Reinforcement Learning and Demonstrations. arXiv preprint arXiv:1709.10087, 2017.

---

> > > > ### Comment · Reviewer_UkXA · 2025-08-06
> > > >
> > > > Thank you again for the detailed rebuttal.
> > > >
> > > > The distinction between the strict guarantees offered by the Projected method and the flexibility-efficiency trade-offs in PD/ALM is now much clearer.
> > > >
> > > > For Q6, can you detail what you get from the experimental data? Is there an analytical result?
> > > >
> > > > The clarification on how your method differs from prior CBF literature should be included in the paper.
> > > >
> > > > It would be helpful to include a diagram or short section summarizing the trade-offs between the different methods.

---

> > > > > ### Author Response · Authors · 2025-08-06
> > > > > **Response to the reviewer's comment: Part I**
> > > > >
> > > > > Thank you very much for your further feedback and valuable suggestions on our work. We are pleased that our previous response clarified the key differences. We will address your follow-up questions in two parts.
> > > > >
> > > > > **Q: Detail what do we get from the data. Is there an analytical result?**
> > > > >
> > > > > R: According to all experimental results, our three constrained diffusion methods—Projected, PD, and ALM—together with SafeDiffuser, outperform Conditional Diffusion in handling strict mathematical constraints. However, their performance varies across different environments. For static imitation and obstacle-avoidance tasks, such as Maze2D, all constrained diffusion methods perform remarkably well in the planning phase, achieving near-zero violations. Both SafeDiffuser and the Projected method can strictly satisfy all constraints, achieving zero violations, because they solve a QP problem at every diffusion step. This strict enforcement, however, comes at a substantial computational cost. In contrast, PD and ALM use numerical iterative updates, which introduce minor violations in the planning phase but still reduce constraint violations by over 99% compared to vanilla Diffuser, while only increasing computation time by approximately 10%. ALM adds a squared constraint penalty, making it slightly more effective in violation reduction than PD at the cost of 10% additional computation, which is still only 1/10 of the closed-form method and 1/100 of QP-based methods.
> > > > >
> > > > > For locomotion tasks, the online performance depends on IDM accuracy. In Hopper, the Projected method again achieves perfect constraint satisfaction during planning but suffers from reward drops during online execution due to its larger deviation from expert trajectories. Occasional online violations still appear because of IDM’s generalization errors and stochasticity, though violations are reduced by 90% relative to Diffuser. In comparison, PD and ALM achieve 96% violation reduction in planning, and during online execution, their looser constraint satisfaction keeps state distributions closer to the expert trajectories, resulting in higher rewards and 85% violation reduction, all while requiring only 1/400 of the computation time of QP with full-trajectory CBF constraints. In Swimmer, where the Projected method achieves only 50% violation reduction online, while PD and ALM reduce violations by 98%. We attribute this to the fact that states generated by PD and ALM are more aligned with the expert distribution, which allows IDM to produce safer actions in execution. Similarly, in SafetyBallRunning, which features dynamic constraints, the Projected method reduces violations by only 50%, with 170% more computation time based on the closed-form projection. In contrast, PD and ALM achieve 99% reduction, with 35% and 119% computation overhead, respectively, showing superior overall performance in this scenario.
> > > > >
> > > > > Providing an analytical results of the computation time and constraint satisfaction for these three methods is always intractable for general cases. This is because the specific form depends on the characteristics of the problem (e.g., the type of constraints, the properties of the expert trajectories distribution, the diffusion model and IDM). We can provide a general analytical framework to help understand these tradeoff. To explore the trade-off, we can decompose the problem into two key factors: the first is the computation time $T$, including the number of iterations $k$ and the per-iteration computational time $t$; and the second is constraint Violation $V$ that measures how far the solution is from satisfying the constraints. For the Projected method, $V = 0$ are strictly satisfied at each iteration, so the number of iterations is diffusion steps. The per-iteration computational time $t$ dominated by the projection step, whose complexity depends on the type of constraints where simple box constraints is $O(n)$ while complex constraints solving a quadratic program could be $O(n^2)$ or $O(n^3)$. For PD and ALM methods, $V$ decreases gradually with the number of iterations $k$, usually following a convergence rate such as $V \sim \frac{1}{k}$ or $V \sim \frac{1}{\sqrt{k}}$ based on some assumptions on the properties of the distribution and constraints. [6] gives a detailed analysis of the sublinear convergence rate in expectation for primal dual methods with respect to the KL divergence in convex case. Therefore, to achieve a desired violation level $V_{\text{desired}}$, the required iterations $k$ can be estimated from the convergence rate (e.g. $k \geq 1 / V_{\text{desired}}$). For per-iteration computational time, the complexity is related to the computation of gradient which is lower than the computational complexity of solving a Quadratic Programming problem.
> > > > >
> > > > > [6] Chamon, Luiz F., Mohammad R. Karimi, and Anna Korba. "Constrained sampling with primal-dual Langevin Monte Carlo." Advances in Neural Information Processing Systems 37 (2024): 29285-29323.

---

> > > > > > ### Comment · Reviewer_UkXA · 2025-08-06
> > > > > >
> > > > > > Thank you I appreciate this additional input -- it would be great to add it to the paper.

---

> > > > > ### Author Response · Authors · 2025-08-06
> > > > > **Response to the reviewer's comment: Part II**
> > > > >
> > > > > **The clarification on how your method differs from prior CBF literature should be included in the paper.**
> > > > >
> > > > > R: Thank you for this valuable suggestion. We completely agree that this is a crucial point. In the revised manuscripts, we will expand our literature review to include a more detailed discussion and comparison with the CBF-related works mentioned during the rebuttal process. This comparison will emphasize our diffusion model's exceptional ability to capture multimodal distributions of safe expert trajectories. Moreover, by leveraging the iterative diffusion process with efficient, gradient-based guidance, our approach enforces safety constraints while avoiding the expensive, per-step Quadratic Programming (QP) optimizations that are common in other CBF methods.
> > > > >
> > > > > **It would be helpful to include a diagram or short section summarizing the trade-offs between the different methods.**
> > > > >
> > > > > R: Thank you for this constructive suggestion. We will include a summarized diagram and section based on the following table.
> > > > > |Method|Constraints Satisfaction|Computational Efficiency|Application|
> > > > > |-|-|-|-|
> > > > > |Projected|Strict zero-violation guarantees; highest safety|Low efficiency; slow with complex constraints|Safety-critical systems requiring strict constraint enforcement|
> > > > > |Primal-Dual (PD)|Small residual violations; weaker satisfaction|Extremely fast; top efficiency|Real-time apps where speed > perfect constraints|
> > > > > |Augmented Lagrangian (ALM)|Strong satisfaction; near-zero violations|High efficiency; slightly slower than PD|General use with balanced speed and safety|
> > > > >
> > > > > We hope this response resolves your concerns.

---

> > > > > > ### Comment · Reviewer_UkXA · 2025-08-06
> > > > > >
> > > > > > Yes this is a nice way to summarize the properties of the 3 models

---

> > > > > > > ### Author Response · Authors · 2025-08-07
> > > > > > > **Response to the reviewer's comment**
> > > > > > >
> > > > > > > Thank you very much for your kind feedback. We are pleased to hear that the summary of the three models' properties is helpful, and we will incorporate this additional input into the revised version of the paper.
> > > > > > >
> > > > > > > Please let us know if there are any remaining concerns regarding updating the final rating. We would be happy to address them.

---

> > > > > > > ### Author Response · Authors · 2025-08-09
> > > > > > > **Response to the reviewer's comment**
> > > > > > >
> > > > > > > We again thank the reviewer for the valuable suggestions to improve our work. We will incorporate the constructive points you raised into the revised paper. We hope our revisions will address the concerns raised and will make the contributions and methodology of our work clearer. We look forward to your final assessment of our manuscript and welcome any additional suggestions you may have for further improvement of the paper.

---

> > > > > > ### Comment · Reviewer_SjPT · 2025-08-07
> > > > > > **Comparison Table: How do traditional QPs play into the picture?**
> > > > > >
> > > > > > > It would be helpful to include a diagram or short section summarizing the trade-offs between the different methods.
> > > > > > > R: Thank you for this constructive suggestion. We will include a summarized diagram and section based on the following table.
> > > > > >
> > > > > > The reviewer very much appreciates the addition of this informative table comparing the various methods. However, I am curious how traditional solving of QPs with CBFs, such as those employed in SafeDiffuser, using, for example, primal-dual interior point methods, plays into the picture? Could you please update the comparison table with a row that compares the methods with traditional QP with CBF solving?

---

> > > > > > > ### Author Response · Authors · 2025-08-08
> > > > > > > **Response to Reviewer SjPT's comment**
> > > > > > >
> > > > > > > Thank you for this suggestion, which allows us to clarify a fundamental point about our contribution.
> > > > > > >
> > > > > > > Firstly, we clarify that according to the codes, SafeDiffuser solved the QP via qpth python package, which uses primal-dual interior point methods and then the solution as the output at each single diffusion step.
> > > > > > >
> > > > > > > To further clarify for table comparison, both our work and SafeDiffuser address the same core problem: a standard diffusion model used for imitation learning does not inherently handle strict mathematical constraints.
> > > > > > >
> > > > > > > SafeDiffuser interprets the diffusion process from a control-theory viewpoint. Each denoising step is treated as a "control action" over the diffusion time horizon. A Control Barrier Function (CBF) constraint is imposed for each intermediate diffusion step, and a full QP is solved at every single step to ensure the resulting sample remains within this safe set.
> > > > > > >
> > > > > > > Our Approach views the diffusion process as a constrained optimization problem. We use our three proposed methods (Projected, PD, ALM) to adjust each diffusion step such that the final trajectory satisfies the constraints. Our key contribution here is showing how efficient constraints gradient-based methods (PD and ALM) can achieve this without the high computational cost of a per-step QP.
> > > > > > >
> > > > > > > Therefore, the main distinction is not about how to solve a QP with a CBF, but about the high-level strategy for making a diffusion-generated trajectory safe.
> > > > > > >
> > > > > > > Furthermore, for online implementation, we specifically use a CBF defined over the physical world trajectory to ensure the final plan is dynamically consistent and safe. SafeDiffuser, in contrast, enforces a hard state constraint at each diffusion time step t.
> > > > > > >
> > > > > > > Our advantage along with SafeDiffuser over other traditional control methods is that we do not require a known objective function; also, for other traditional QP safety filter with CBF, they usually have additional assumptions, including that the system should be control affine, which restricts the applicability. Our methods leverage the powerful multimodal modeling capabilities of diffusion models to imitate complex expert behaviors and the properties of diffusion process to handle the constraints.
> > > > > > >
> > > > > > > For these reasons, we believe the most direct and informative comparison is separating SafeDiffuser from other traditional methods with CBF, as this allows for a fair evaluation of two frameworks for safe diffusion and others.  We have our new rows as follows:
> > > > > > > |Method|Constraints Satisfaction|Computational Efficiency|Application|
> > > > > > > |-|-|-|-|
> > > > > > > |SafeDiffuser|Higher constraint satisfication|Low efficiency; Solving QP with CBF at each step|Safety-critical systems requiring strict constraint enforcement|
> > > > > > > |Traditional QP with CBF|Strict constraint satisfication|Average efficiency; Solving QP with CBF|Known objective functions, control affine systems|
> > > > > > >
> > > > > > > We will add this part in the revised paper and hope this response resolves your concerns.

---

> > > > > > > > ### Comment · Reviewer_SjPT · 2025-08-08
> > > > > > > > **Re. Table with Comparison to Safe Diffuser**
> > > > > > > >
> > > > > > > > > For these reasons, we believe the most direct and informative comparison is separating SafeDiffuser from other traditional methods with CBF, as this allows for a fair evaluation of two frameworks for safe diffusion and others. We have our new rows as follows:
> > > > > > > >
> > > > > > > > The reviewer appreciates the more comprehensive comparison to SafeDiffuser and the differences are (almost) fully clear now.
> > > > > > > >
> > > > > > > > However, this is quite an intricate topic with the differences between the proposed method and SafeDiffuser really lying in the details. Therefore, I strongly suggest that the authors carefully think about how to both clearly and fairly present the differences in the revised paper.
> > > > > > > >
> > > > > > > > Furthermore, I suggest that you separate in the comparison table the discussion with respect to "Planning" and "Inference". This might make it easier to accurately track the differences across those two settings.
> > > > > > > >
> > > > > > > > In summary, the reviewer is satisfied with the responses from the authors.

---

> > > ### Author Response · Authors · 2025-08-06
> > > **Follow-up on Rebuttal**
> > >
> > > Dear Reviewer,
> > >
> > > Thank you again for your time and for the insightful suggestions that helped us improve our work.
> > >
> > > Regarding your comment about the limitations section, we would like to clarify that, due to space constraints, it is included in the appendix, which is provided in the supplementary materials and packaged with our code as a single ZIP file. If there are any remaining access issues, we would be happy to provide the appendix text directly during the discussion phase.
> > >
> > > As the discussion period is coming to a close, we would like to kindly check whether our previous response addressed your concern, or if you have any further points you would like to discuss. We are happy to provide any additional clarifications if needed.
> > >
> > > Best regards,
> > >
> > > Authors

---

> > ### Comment · Reviewer_SjPT · 2025-08-07
> > **Comparison with SafeDiffuser**
> >
> > I am a bit puzzled and confused by some of the author's comments in their rebuttal.
> >
> > > Compared to SafeDiffuser, our key difference is that it applies CBFs at intermediate diffusion steps, whereas we enforce CBFs along the entire trajectory in the physical world
> >
> > - When I read Section 4.1 about the "Projected Method", it is applied at each denoising step (i.e., at each intermediate diffusion step). How is this a difference then?
> > - Also, does SafeDiffuser not enforce CBFs along the entire trajectory?
> >
> > > Our gradient-based constraint enforcement is also more efficient than typical QP-based safety layers.
> >
> > Can you please explain why this is? Also, please be more specific. Do you explicitly mean that the "Projected Method" is more efficient than QP-based safety layers? Why is this the case? And why is then your/the "Projected Method" not typically used to solve QP-based safety layers?

---

> > > ### Author Response · Authors · 2025-08-08
> > > **Response to Reviewer SjPT's comment II**
> > >
> > > We thank the reviewer for the insightful questions to improve our paper.
> > >
> > > > When I read Section 4.1 about the "Projected Method", it is applied at each denoising step (i.e., at each intermediate diffusion step). How is this a difference then?
> > >
> > > > Also, does SafeDiffuser not enforce CBFs along the entire trajectory?
> > >
> > > You are right that our Projected method also applies a correction at every diffusion step. The key distinction is how it is formulated and computed, which stems from a fundamental difference in approach.
> > >
> > > Both our method and SafeDiffuser aim to ensure the generated trajectory remains within a safe set defined by: $h(x^\tau) \ge 0,\forall\tau\in\\{0,\dots,\mathcal{T}\\}$. As mentioned in the previous comment, SafeDiffuser views the diffusion process from a control perspective, treating the update at each diffusion step t as a control action u. To ensure safety, it solves CBF-QP at each diffusion step: $\min_u \\|u_t-u_t^ *\\|^2 \quad \text{s.t.} \quad \dot{h(x_t^\tau)}u_t+\alpha h(x_t^\tau)\ge 0, \forall\tau\in\\{0,\dots,\mathcal{T}\\}$ where the gradient of $h(\cdot)$ is based on the diffusion dynamics with $t$ rather than physical environment dynamics with $\tau$.
> > >
> > > Our Projected Method is inspired by projected gradient descent and projected Langevin dynamics。 We perform a different operation that directly project these sampling points onto the feasible set: $\min_{x} \\| x_t - x_t^ * \\| \quad \text{s.t.}\quad h(x_t^\tau) \ge 0, \forall\tau\in\\{0,\dots,\mathcal{T}\\}.$ We directly use the state constraint defined by the safe set.
> > >
> > > Then based on all these constrained diffuser framework, we later apply a DCBF as our safe set which is defined over the entire physical world trajectory: $h(x^{\tau+1}) \ge (1 - \alpha) h(x^\tau), \quad \forall\tau\in\\{0,\dots,\mathcal{T}\\}.$ And also, in our understanding, SafeDiffuser does not apply a CBF to the physical trajectory in this manner.
> > >
> > > Ours is to make sure the trajectory is forward invariant, but safe diffuser is trying to make the diffusion process forward invariant, and both of these two methods aim and success to achieve safety of the final planned trajectory, but with different ways of using CBF and therefore different problem formulation and detailed update rules.
> > >
> > > > Explanation on "our gradient-based constraint enforcement is also more efficient than typical QP-based safety layers."
> > >
> > > We thank the reviewer for pointing out the lack of precision in our earlier statement. To clarify, by gradient-based methods we specifically refer to approaches that handle constraints through the use of gradients of constraints, i.e., PD and ALM.
> > >
> > > Regarding projection-based methods, you are correct: in practice, the projection step is formulated as a Quadratic Program (QP) that is solved at each diffusion step. Compared to SafeDiffuser, our method does not require computing the gradient of the constraints with respect to the diffusion dynamics. This makes the constraint formulation simpler. However, because the projection still typically requires solving a QP (not limited to QP) at each step, the computational cost is not inherently more efficient in the general case. In cases where the constraints admit a closed-form projection operator (for example, simple convex sets such as Euclidean balls, half-spaces, or box constraints), the projection step can be performed analytically without solving a QP. In these cases, the computational cost is significantly reduced.
> > >
> > > We will include this clarification in the revised paper.

---

> > > > ### Comment · Reviewer_SjPT · 2025-08-08
> > > > **Difference in Safety Constraints between Planning and Inference for your Method**
> > > >
> > > > > Our Projected Method is inspired by projected gradient descent and projected Langevin dynamics。 We perform a different operation that directly project these sampling points onto the feasible set: $\min_{x} \| x_t - x_t^ * \| \quad \text{s.t.}\quad h(x_t^\tau) \ge 0, \forall\tau\in\{0,\dots,\mathcal{T}\}.$ We directly use the state constraint defined by the safe set.
> > > > >
> > > > > Then based on all these constrained diffuser framework, we later apply a DCBF as our safe set which is defined over the entire physical world trajectory: $h(x^{\tau+1}) \ge (1 - \alpha) h(x^\tau), \quad \forall\tau\in\{0,\dots,\mathcal{T}\}.$ And also, in our understanding, SafeDiffuser does not apply a CBF to the physical trajectory in this manner.
> > > > >
> > > > > Ours is to make sure the trajectory is forward invariant, but safe diffuser is trying to make the diffusion process forward invariant, and both of these two methods aim and success to achieve safety of the final planned trajectory, but with different ways of using CBF and therefore different problem formulation and detailed update rules.
> > > >
> > > > This difference in the formulation of the constraints between planning (where you only use state-based constraints) and inference (where you use CBFs that are aware of the actuation dynamics) was not clear to me when initially reading the paper. When you revise the paper, please ensure that it becomes more obvious and easier to understand how you differently define constraints for each stage and what are the reasons/motivations for that.
> > > >
> > > > The reviewer is satisfied with the author's response.

---

> > > > > ### Author Response · Authors · 2025-08-09
> > > > > **Response to Reviewer SjPT's comment**
> > > > >
> > > > > Thank you very much for your positive feedback and constructive suggestions.
> > > > >
> > > > > We agree that the distinctions you've highlighted are important. We will carefully revise the manuscript according to your recommendations to improve the paper's clarity, especially concerning the comparison with SafeDiffuser and the different constraint formulations for the planning and inference stages.
> > > > >
> > > > > Thank you again for your valuable input.

---

### Note · Authors · 2025-08-13

We thank the area chair for their efforts in managing the review process, and all reviewers for their time and for providing thoughtful and valuable feedback. We are glad that the reviewers recognized the contributions of our work and its value to diffusion-based safe planning and control, particularly in addressing the critical challenge of enforcing safety without retraining and connecting to constrained optimization. The feedback provides us with a clear direction for strengthening the paper, and we will incorporate the reviewers’ suggestions and address all raised concerns in the revision.

In particular, we will refine the introduction and contribution statement to present a clearer overview of our framework, supported by an illustrative figure showing how our methods modify the reverse diffusion process to enforce constraints. The discussion in the related work section will be expanded with a table comparing our approach to key prior work, such as SafeDiffuser and CBF-based planning and control methods. We will provide a detailed quantitative analysis of trade-offs among our methods, and add new experiments, including an online comparison with SafeDiffuser augmented with our IDM to ensure state-action consistency, a worst-case analysis in the PybulletBallRunning task under challenging initial conditions, an evaluation on high-dimensional dexterous hand manipulation, a sensitivity analysis on IDM inaccuracies and safety performance, and an ablation study on executing multiple time steps before replanning. We will also clarify points regarding IDM and discuss about practical considerations such as actuation constraints, along with other revisions suggested by the reviewers.

We believe these updates will resolve all of the reviewers’ concerns and make the contributions, methodology, and practical implications of our work much clearer. We hope you will take these improvements into account in your final evaluation. Thank you again for your time and efforts.

---

### Decision · Program_Chairs · 2025-09-17

**Decision:**

Accept (poster)

**Comment:**

This paper introduces Constrained Diffusers, a novel framework for incorporating safety constraints into pretrained diffusion models used for planning and control. The key innovation is the development of three constrained sampling methods, which guide the reverse diffusion process to produce constraint-satisfying trajectories without retraining or modifying the model architecture. Additionally, the authors propose using discrete control barrier functions to ensure dynamic feasibility and safety during online execution. The framework is evaluated on standard and high-dimensional simulated tasks, showing improved safety and competitive task performance compared to baselines like SafeDiffuser and Conditional Diffuser.

The paper’s strengths lie in its timely and impactful contributions at the intersection of diffusion models and safety-critical control. It provides a modular, efficient alternative to retraining-heavy or optimization-intensive methods, and the three proposed constrained sampling methods exhibit clear trade-offs between safety and computational efficiency. The paper demonstrates broad applicability, including to high-dimensional tasks, and offers insightful comparisons to prior work. The rebuttal added new empirical studies (e.g., worst-case analysis, robustness to IDM errors, and high-dimensional scenarios) that significantly strengthened the paper.

The main weaknesses were related to clarity, presentation of theoretical concepts, and the need for more explicit comparisons to related work (especially SafeDiffuser). Some reviewers also noted that the original manuscript lacked a clear discussion of limitations and constraints on generalizability. These concerns were effectively addressed during the rebuttal, with the authors providing detailed clarifications, new results, and plans to revise the manuscript significantly for clarity and completeness.

In conclusion, this paper presents a novel, practical, and well-validated contribution to safe planning with diffusion models. While not perfectly polished in its initial form, the authors' responsiveness and the strength of the core ideas warrant acceptance at NeurIPS. I encourage the author to carefully revise their paper based on reviewers' comments.